# Inverse modeling of SO$_2$ and NO$_x$ emissions over China using multi-sensor satellite data: 1. formulation and sensitivity analysis

Yi Wang[1], Jun Wang[1,2], Xiaoguang Xu[2,3], Daven K. Henze[4], Zhen Qu[4], Kai Yang[5]

[1]Interdisciplinary Graduate Program in Informatics, The University of Iowa, Iowa City, IA 52242, USA
[2]Department of Chemical and Biochemical Engineering, and Center for Global & Regional Environmental Research, The University of Iowa, Iowa City, IA 52242, USA
[3]Joint Center for Earth Systems Technology and Department of Physics, University of Maryland Baltimore County, Baltimore, Maryland, 21250, USA
[4]Department of Mechanical Engineering, University of Colorado Boulder, Boulder, CO 80309, USA
[5]Department of Atmospheric and Oceanic Sciences, University Maryland, College Park, MD 20742, USA

*Correspondence to*: Jun Wang (jun-wang-1@uiowa.edu) and Yi Wang (yi-wang-4@uiowa.edu)

**Abstract.** SO$_2$ and NO$_2$ observations from the Ozone Mapping and Profiler Suite (OMPS) sensor are used for the first time in conjunction with GEOS-Chem adjoint model to optimize both SO$_2$ and NO$_x$ emission estimates over China for October 2013. Separate and joint (simultaneous) optimizations of SO$_2$ and NO$_2$ emissions are both conducted and compared. Posterior emissions, compared to the prior, yield improvements in simulating columnar SO$_2$ and NO$_2$, in comparison to measurements from OMI and OMPS. The posterior SO$_2$ and NO$_x$ emissions from separate inversions are 748 Gg S and 672 Gg N, which are 36% and 6% smaller than prior MIX emissions (valid for 2010), respectively. In spite of the large reduction of SO$_2$ emissions over the North China Plain, the simulated sulfate-nitrate-ammonium Aerosol Optical Depth (AOD) only decrease slightly, which can be attributed to (a) nitrate rather than sulfate as the dominant contributor to AOD and (b) replacement of ammonium sulfate with ammonium nitrate as SO$_2$ emissions are reduced. For joint inversions, both data quality control and the weight given to SO$_2$ relative to NO$_2$ observations can affect the spatial distributions of the posterior emissions. When the latter is properly balanced, the posterior emissions from assimilating OMPS SO$_2$ and NO$_2$ jointly yield a difference of -3% to 15% with respect to the separate assimilations for total anthropogenic SO$_2$ emissions and ±2% for total anthropogenic NO$_x$ emissions; but the differences can be up to 100% for SO$_2$ and 40% for NO$_2$ in some grid cells. Improvements on SO$_2$ and NO$_2$ simulations from the joint inversions are overall consistent with those from separate inversions. Moreover, the joint assimilations save ~50% of the computational time than assimilating SO$_2$ and NO$_2$ separately in a sequential manner of computation. The sensitivity analysis shows that a perturbation of

NH$_3$ to 50% (20%) of the prior emission inventory: (a) has negligible impact on the separate SO$_2$ inversion, but can lead to decrease of posterior SO$_2$ emissions over China by -2.4% (-7.0%) in total and up to -9.0% (-27.7%) in some grid cells in the joint inversion with NO$_2$; (b) yield posterior NO$_x$ emissions over China decrease by -0.7% (-2.8%) for the separate NO$_2$ inversion and by -2.7% (-5.3%) in total and up to -15.2% (-29.4%) in some grid cells for the joint inversion. The large reduction of SO$_2$ between 2010 and 2013, however, only leads to ~10% decrease of AOD regionally; reducing surface aerosol concentration requires the reduction of emissions of NH$_3$ as well.

## 1. Introduction

Both SO$_2$ and NO$_2$ in the atmosphere have adverse impacts on human health and can affect radiative forcing that leads to climate change. Not only do they cause inflammation and irritation of the human's respiratory system, but they also react with other species to form sulfate and nitrate aerosols (Seinfeld and Pandis, 2016), which subsequently can lead to or exacerbate respiratory and cardiovascular diseases (Lim et al., 2012). Sulfate and nitrate account for the largest mass of anthropogenic aerosols, which contributed to ~3 million premature deaths worldwide in 2010 (Lelieveld et al., 2015). In addition to the health impacts, anthropogenic sulfate and nitrate particles are estimated to have caused -0.4 and -0.15 W m$_{-2}$ radiative forcing, respectively, on a global scale between 1750 and 2011 through scattering solar radiation, and via modifying cloud microphysical properties (Myhre et al., 2013).

Satellite-derived global distributions of SO$_2$ and NO$_2$ Vertical Column Densities (VCDs) have been used to study the aforementioned impacts of SO$_2$ and NO$_2$ on atmospheric composition, climate change, and human health. In particular, since SO$_2$ and NO$_2$ VCDs are, to first order, linearly related to SO$_2$ and NO$_x$ emissions (Calkins et al., 2016), they can be used to update bottom-up emission inventories that have large uncertainties and a temporal lag often of at least one year (Liu et al., 2018). Of particular interest for this study is China, which has large SO$_2$ and NO$_x$ emissions from anthropogenic sources (coal-fired power plants, industry, transportation, and residential activity). Moreover, China has seen a 62% reduction in anthropogenic SO$_2$ emissions and a 17% reduction of anthropogenic NO$_x$ emissions on average from 2010 to 2017 (Zheng et al., 2018) due to the implementation of emission control policies, and these changes vary by regions, cities (Liu et al., 2016), and sectoral sources (Zheng et al., 2018). The reduction of SO$_2$ emissions mainly occurred in the coal-fired power plants and industries while the decrease of NO$_x$ emissions was largely ascribed to coal-fired power plants (Zheng et al., 2018). Noticeable uncertainties larger than 30% for both anthropogenic SO$_2$ and NO$_x$ in 2010 over China were documented (Li et

al., 2017b) and can be larger at the regional scale due to the uncertainty of activity rates, emission factors, and spatial proxies, which are used in the bottom-up approach (Janssens-Maenhout et al., 2015). The uncertainty is large and can be compounded by possible discrepancies caused by the temporal lag of bottom-up emission inventories and the rapid changes of emissions over time.

Several methods have been developed to update $SO_2$ and $NO_x$ emissions using satellite VCD retrievals of $SO_2$ and $NO_2$, which have global coverage and allows near-real-time access. The mass balance method, which scales prior emissions by the ratios of observed VCDs to Chemistry Transport Model (CTM) counterparts, was applied to $SO_2$ retrievals from SCanning Imaging Absorption SpectroMeter for Atmospheric CHartographY (SCIAMACHY) and Ozone Monitoring Instrument (OMI) (Lee et al., 2011;Koukouli et al., 2018) and to $NO_2$ from Global Ozone Monitoring Experiment (GOME) and OMI (Martin et al., 2003;Lamsal et al., 2010) to estimate $SO_2$ and $NO_x$ emissions, respectively. Lamsal et al. (2011) simulated the sensitivity of VCDs to emissions (the finite difference mass balance approach) using a CTM, which was applied to OMI $NO_2$ retrievals to estimate $NO_x$ emissions. $SO_2$ VCD retrievals from GOME, GOME-2, SCIAMCHY, and Ozone Mapping and Profiler Suite (OMPS) were used to estimate point sources through linear regression between VCDs and emissions or function fitting, although the method can only detect about half of the total anthropogenic $SO_2$ emissions (Li et al., 2017a;Zhang et al., 2017;Fioletov et al., 2013;Fioletov et al., 2016). With explicit considerations of chemistry, transport, and deposition, the four-dimension variational data assimilation (4D-Var) approach was applied to estimate emissions using $SO_2$ data from OMI (Wang et al., 2016;Qu et al., 2019a), and $NO_2$ data from SCIAMCHY, GOME-2, and OMI (Kurokawa et al., 2009;Qu et al., 2017;Kong et al., 2019). The 4D-Var posterior has a smaller root mean square error than the mass balance posterior, especially in the conditions when the initial guess and true emissions have different spatial patterns (Qu et al., 2017); this is because the spatial extent of source influences on modelled column concentrations (Turner et al., 2012) are only indirectly and partially accounted for in the mass balance approach. Cooper et. al (2017), however, showed that the iterative finite difference mass balance approach has similar normalized mean error value as the 4D-Var approach for global-scale models with coarse resolution when synthetic $NO_2$ columns observations are used to constrain $NO_x$ emissions. To combine the strengths of the 4D-Var and mass balance approaches, Qu et al. (2017) further introduced a hybrid 4D-Var-mass-balance approach, which can better capture trends and spatial variability of $NO_x$ emissions than the mass balance approach and save significant computational resources when applied to constrain monthly $NO_x$ emissions for multiple years. Other data assimilation approaches including the ensemble Kalman

filter method (Miyazaki et al., 2012;Miyazaki et al., 2017) and the Daily Emission estimates Constrained by Satellite Observation (DECSO) algorithm (Mijling and van der A, 2012;Ding et al., 2015) have also been used to constrain $NO_x$ emissions.

Here, we focus on the development and feasibility for joint 4D-var assimilation of satellite-based $SO_2$ and $NO_2$ data to optimize $SO_2$ and $NO_x$ emission strengths simultaneously. Specifically, this study aims to conduct 4D-Var assimilation of VCDs of $SO_2$ and $NO_2$ from OMPS to constrain $SO_2$ and $NO_x$ emissions over China using the GEOS-Chem 4D-Var inverse modeling framework. In our companion study (Wang et al., 2019), we develop approaches to downscaling the optimized emission inventories for improving air quality predictions. Despite their numerous applications for top-down estimate of $SO_2$ and $NO_x$ emissions in the past two decades, GOME and SCIAMCHY stopped providing data in 2004 and 2012, respectively, while OMI has been suffering from a row anomaly that leads to much less spatial coverage and larger data uncertainty (Schenkeveld et al., 2017). Hence, it is important to study the potential of next-generation sensors such as OMPS toward continuously monitoring the change of $SO_2$ and $NO_x$ emissions and their atmospheric loadings. Two OMPS sensors onboard Suomi NPP and NOAA-20 have been launched in 2011 and 2018, respectively, and the third one is expected to be launched in 2020. As OMPS will continue to provide $SO_2$ and $NO_2$ retrievals in the next two decades, this study, for the first time, seeks to provide a critical assessment of the extent to which the OMPS observations improve emissions estimates and air quality forecast at the regional scale.

The novelty of this study lies not only in the first application of OMPS $SO_2$ and $NO_2$ retrievals to constrain emissions using the 4D-Var technique but also in the deployment of OMI data to assess the GEOS-Chem simulation with posterior emissions, thereby studying the degree to which OMPS and OMI retrievals, despite their difference in sensor characteristics and inversion techniques, can provide consistent constraints for the model improvement. Qu et al. (2019a) showed that posterior $SO_2$ emissions derived from different OMI $SO_2$ products vary in strength and have consistent trend signs. Our study here using OMPS thus examines an important issue, which is whether or not there would be any artificial trends in our climate data record of atmospheric $SO_2$ and $NO_2$ due to the transition of satellite sensors (Wang and Wang, 2020). Our study is also different from past studies (Wang et al., 2016;Qu et al., 2017;Qu et al., 2019a;Qu et al., 2019b) that have applied the 4D-Var technique to OMI data with the GEOS-Chem adjoint model, but did not include evaluation with independent satellite data. Qu et al. (2019b) showed joint inversion using OMI $SO_2$ and $NO_2$ benefits from simultaneous adjustment of OH and $O_3$ concentrations, which supports assimilating OMPS $SO_2$ and $NO_2$ observations simultaneously in this study.

Additionally, considering that the uncertainty of NH$_3$ emission inventories is up to 153% over China (Kurokawa et al., 2013) and NH$_3$ emissions are not constrained in our inversions, we also explore issues related to the co-variation among species that appear to be independent but indeed are connected through chemical processes and analyze the differences in responses of emissions and aerosols to NH$_3$ emissions uncertainty between joint and single-species assimilations. Finally, this paper also provides the foundation for the Part II investigation (Wang et al., 2019) in which we develop various downscaling methods to illustrate that optimized emission, albeit its coarse resolution inherent from OMPS data, can be used to improve the air quality forecast at the resolution much finer than OMPS pixel size.

We describe OMPS and OMI data in Sect. 2. The GEOS-Chem model and its adjoint as well as the design of numerical experiments are presented in Sect. 3. Results of case studies for October 2013 are provided in Sect. 4. Sect. 5 consists of discussion and conclusions.

## 2. Data

### 2.1 OMPS data as constraints

We use OMPS Level-2 SO$_2$ and NO$_2$ tropospheric VCDs in October 2013 as constraints to optimize SO$_2$ and NO$_x$ emissions over China. The OMPS nadir mapper on board the Suomi-NPP satellite, launched in November 2011, observes hyperspectral solar radiance and earthshine radiance at 300-380 nm (Flynn et al., 2014). With 35 detectors of 50x50 km nominal pixel size in cross-track direction, OMPS has a swath of 2800 km flying across the equator at 1:30 PM local time ascendingly at the sunlit side of the Earth surface and providing global coverage daily. Both SO$_2$ and NO$_2$ are retrieved through the Direct Vertical Column Fitting (DVCF) algorithm with SO$_2$ and NO$_2$ atmospheric profile information from GEOS-Chem simulations and have a retrieval precision of 0.2 DU and 0.011 DU, respectively, which are estimated from the standard background (a clean region that is far from emission sources) retrievals (Yang et al., 2013;Yang et al., 2014). These precision values can be used as the observation error in the cost function of data assimilation. However, we should notice that the estimated observation (retrieval) errors only represent the observation error distribution of the products as a whole; it cannot represent the observation error distribution for every pixel, because the pixel-level error is amenable to spatiotemporal change of cloud fraction, satellite observation geometry, aerosol impacts, etc. In theory, if the uncertainties can be analytically described at the pixel level, they would be directly applied to improve the satellite product in the first place.

Only pixels with both Solar Zenith Angle (SZA) and View Zenith Angle (VZA) less than 75° are used, as larger SZA or VZA result in longer light path length, and consequently less information content and lower data quality for retrieving the change of $SO_2$ or $NO_2$ loadings in the Plane Boundary Layer (PBL) where the two trace gases from anthropogenic sources mainly concentrate. We also remove the pixels with Radiative Cloud Fraction (RCF) larger than 0.2 for $SO_2$ and 0.3 for $NO_2$ as a trade-off between the data amount and cloud impacts. Considering their large uncertainty, OMPS $SO_2$ retrievals in the grid cell where the prior simulation is less than 0.1 DU will not be used, except in Quality Control (QC) sensitivity analysis experiments.

**2.2 OMI data for assessment**

OMI Level-3 $SO_2$ and $NO_2$ tropospheric VCDs at a spatial resolution of 0.25°x0.25° from NASA are used for evaluating the model results. OMI is a UV-vis hyperspectral sensor that observes solar irradiance and earthshine radiance at 300-500 nm. The swath of OMI is 2600 km, consisting of 60 detectors with the nominal pixel size of 13x24 $km_2$ at nadir. OMI flies across the equator in the ascending node at 1:45 PM local time, which is very close to the 1:30 PM local time for OMPS. Due to row anomaly (Schenkeveld et al., 2017), OMI takes more than one day to provide global coverage. The Level-3 product is derived from the Level-2 product; the latter is retrieved through the Principal Component Analysis (PCA) algorithm with a fixed Air Mass Factor (AMF) assumption for $SO_2$ (Li et al., 2013) and variation of the Differential Optical Absorption Spectroscopy (DOAS) algorithm for $NO_2$ (Krotkov et al., 2017;Marchenko et al., 2015), with a precision of 0.5 DU (Li et al., 2013) and 0.017 DU (Krotkov et al., 2017), respectively. In the Level-3 product, pixels affected by row anomaly are removed. For $SO_2$, only the pixel with the shortest light path, SZA less than 70°, RCF less than 0.2, and detector number in the range of 2 to 59 (1-based) is retained in a 0.25°x0.25° grid cell and then corrected with a new AMF based on GEOS-Chem $SO_2$ profile simulation (Leonard, 2017). For the OMI Level-2 $NO_2$ product, the AMF calculation is based on Global Modeling Initiative $NO_2$ profile simulation (Krotkov et al., 2017), and all pixels with SZA less than 85°, terrain reflectivity less than 0.3, RCF less than 0.3 are averaged in a 0.25°x0.25° grid cell weighted by the overlapping area of grid cell and pixel to form Level-3 product (Bucsela et al., 2016). In the assessments, OMI observations are averaged at 2°x2.5° model grid cell, and model simulations are sampled by OMI observational time.

**3. Method**

### 3.1 GEOS-Chem and its adjoint

GEOS-Chem is a 3-D chemistry transport model driven by emissions and GEOS-FP meteorological fields. The secondary sulfate-nitrate-ammonium aerosol formation in the model is introduced by Park et al. (2004). Both aerosols and gases are removed by wet deposition, including washout and rainout from large-scale or convective precipitation (Liu et al., 2001) and the dry deposition following a resistance-in-series scheme with aerodynamic resistance and boundary resistance calculated from GEOS-FP meteorological field and surface resistances based largely on a canopy model (Wang et al., 1998;Wesely, 1989). Anthropogenic $SO_2$, $NO_x$, and $NH_3$ emissions used over East Asia are the mosaic emission inventory (MIX) (Li et al., 2017b) for year 2010. $SO_2$ and $NO_2$ VCDs are simulated at 2°x2.5° resolution with 47 vertical layers using both the prior and posterior emission inventories to compare with OMI retrievals.

The GEOS-Chem adjoint model is a tool for efficiently calculating the sensitivity of a scalar cost function with respective to large numbers of model parameters simultaneously such as emissions (Henze et al., 2007). In this study, the cost function is defined as Eq. (1).

$$J(\boldsymbol{\sigma}) = \gamma \frac{1}{2}\big[H_{SO2}\big(M(\boldsymbol{\sigma})\big) - \mathbf{c}_{SO2}\big]^T \mathbf{S}_{SO2}^{-1}\big[H_{SO2}\big(M(\boldsymbol{\sigma})\big) - \mathbf{c}_{SO2}\big] + \frac{1}{2}\big[H_{NO2}\big(M(\boldsymbol{\sigma})\big) - \mathbf{c}_{NO2}\big]^T \mathbf{S}_{NO2}^{-1}\big[H_{NO2}\big(M(\boldsymbol{\sigma})\big) - \mathbf{c}_{NO2}\big] + \frac{1}{2}[\boldsymbol{\sigma} - \boldsymbol{\sigma}_a]^T \mathbf{S}_a^{-1}[\boldsymbol{\sigma} - \boldsymbol{\sigma}_a] \quad (1)$$

$\mathbf{E}$ is a vector in which $SO_2$ and $NO_x$ emissions are ordered by GEOS-Chem model grid cell and by species. $\mathbf{E}_a$ is a prior estimate, and $\boldsymbol{\sigma}$ is a state vector, consisting of $\ln(E_i/E_{a,i})$, where $E_i$ and $E_{a,i}$ are the $i_{th}$ element in $\mathbf{E}$ and $\mathbf{E}_a$, respectively. $\mathbf{c}_{SO2}$ and $\mathbf{c}_{NO2}$ are vectors of OMPS $SO_2$ and $NO_2$ tropospheric VCDs, respectively. $\mathbf{S}_{SO2}$ and $\mathbf{S}_{NO2}$ are observation error covariance matrixes for $SO_2$ and $NO_2$ and are assumed to be diagonal, which means observational errors are uncorrelated. M is the GEOS-Chem model that simulates the relationship between $SO_2$ and $NO_2$ concentrations in the atmosphere and the emissions factors. $H_{SO2}$ and $H_{NO2}$ are observation operators which map GEOS-Chem simulations of $SO_2$ and $NO_2$ to the observational space, respectively. $\boldsymbol{\sigma}_a$ is the prior estimate of $\boldsymbol{\sigma}$, and $\mathbf{S}_a$ is the error covariance matrix for $\boldsymbol{\sigma}_a$. $\mathbf{S}_a$ is assumed to be diagonal with a relative error of 50% for $SO_2$ and 100% for $NO_x$ as used in Xu et al. (2013).  $\gamma$ is a parameter we introduce to balance the importance of the $SO_2$ observation term (first term on the right side of Eq. (1)) and $NO_2$ observational term (second term on the right side of Eq. (1)), given both the different sizes and observation errors of these two observation datasets.

OMPS $SO_2$ and $NO_2$ tropospheric VCDs are directly compared to GEOS-Chem tropospheric VCDs of $SO_2$ ($H_{SO2}(M(\boldsymbol{\sigma}))$ in Eq. (1)) and $NO_2$ ($H_{NO2}(M(\boldsymbol{\sigma}))$ in Eq. (1)). Retrieving satellite $SO_2$ and $NO_2$ tropospheric VCDs requires assumptions regarding $SO_2$ and $NO_2$ vertical profiles, as the sensitivity of the radiance observed by satellite sensors to the changes of $SO_2$ or $NO_2$ loadings is a function of plume height. If the vertical profile

 assumptions in the retrieval process are inconsistent with the GEOS-Chem simulations, the inconsistency partly contributes to the difference between the GEOS-Chem simulations and the OMPS retrievals ($H_{SO2}(M(\boldsymbol{\sigma})) - \mathbf{c}_{SO2}$ or $H_{NO2}(M(\boldsymbol{\sigma})) - \mathbf{c}_{NO2}$). In this study, OMPS $SO_2$ and $NO_2$ tropospheric VCDs are retrieved using the shape of vertical profiles from GEOS-Chem simulations (Yang et al., 2013;Yang et al., 2014), but the differences of model version, simulation year, and emission inventory still exist. These differences still can lead to the differences of

 vertical profiles, hence partly contributing to the difference between the GEOS-Chem simulations and the OMPS retrievals. The vertical profile differences can lead to a mean bias of -6.8% overall at the pixel level (Fig. S1) and -7.5% (Fig. S2) for OMPS $SO_2$ and $NO_2$ retrievals, respectively. And we shall discuss the impacts of these bias on emission inverse modeling in Sect. 4.1.1.

 In the optimization formulation, the forward model errors are also considered as part of the observation error term. However, while several ways to construct model error covariance matrix exist, including the Hollingworth-Lönnberg (Hollingsworth & Lönnberg, 1986) and NMC (Bannister, 2008) methods, their application for off-line CTM model error characterization deserves a separate a study. The Hollingsworth method extracts observation error variance (including forward model error) from (observation – background) covariance statistics with the

 assumptions that observation error is spatially uncorrelated, background error is spatially correlated as a function of distance, and observation error and background error are uncorrelated. The assumption that background error is spatially correlated as a function of distance only is suitable for the meteorological fields that vary smoothly, but for chemical species, emissions also contribute significantly to model errors and emissions are spatially correlated. The NMC method is normally applied to weather forecast models or on-line-coupled weather-

 chemistry models (Benedetti and Fisher, 2007). Off-line CTMs such as GEOS-Chem use the meteorological reanalysis and so, NMC is not applicable here to quantify the CTM's transport error. Consequently, CTM's transport errors are neglected in the past emission optimization work (Wang et al., 2016) and are adopted in this study. Admittedly, this simplification should be studied in future together with the evaluation and developments of methods to characterize off-line CTM errors.

We developed the observation operators for OMPS $SO_2$ and $NO_2$, and the validations are shown in Fig. 1. The sensitivities of the cost function with respect to anthropogenic $SO_2$ and $NO_x$ emissions from the adjoint model is consistent with the sensitivities calculated through the finite difference approach. Hence, Fig. 1 confirms the correctness of the new observation operators integrated into the GEOS-Chem adjoint model.


To optimize the emission inventories, $\boldsymbol{\sigma}$ is adjusted iteratively until the cost function is minimized. The minimization is conducted with the L-BFGS-B algorithm (Byrd et al., 1995), which utilizes the sensitivity of the cost function with respect to $\boldsymbol{\sigma}$ that is calculated by the GEOS-Chem adjoint model. The minimization process halts when the difference in the cost function between two consecutive iterations is less than 3%. This selection

is to expedite the computation while still maintain the similar accuracy for the optimization; further tests show that the more iterations (after <3% reduction of cost function) doesn't yield discernable difference in the cost function values (Fig. S3) and optimization results (Table S1 and S2).

### 3.2 Experiment design

Several elements play a role in the inverse modeling of emissions, including data quality control, balancing the

spatial distributions of observational frequencies for the same species, balancing the observation contributions from different species, and uncertainties in the $NH_3$ emission inventory (because $NH_3$ has impacts on $SO_2$ and $NO_2$ lifetimes). To investigate the impacts of these factors on the posterior emissions, we design a set of experiments as summarized in Table 1 and Table 2. All these experiments use OMPS $SO_2$ and $NO_2$ retrievals to optimize corresponding emissions over China in October 2013 at a horizontal resolution of 2°×2.5°. Although

finer resolution options such as 0.5°×0.625° or 0.25°×0.3125° are available for China, the 2°×2.5° resolution is selected for two reasons: (1) Save computational time; (2) chiefly, the coarse resolution of OMPS retrievals (50 km × 50 km at nadir and 190 km x 50 km at edges) has no first-order information to resolve the emissions at fine-resolution of 0.5°×0.625° or 0.25°×0.3125°. In Part II (Wang et al., 2019) of this study, we develop downscaling tools for regional air quality modeling. Indeed, one of the goals of the two-part investigation is to illustrate how

OMPS data could be used to improve the air quality forecast at the resolution much finer than OMPS pixel size.

### 3.2.1 Control experiments

The first control experiment is E-$SO_2$ (Table 1), in which only OMPS $SO_2$ tropospheric VCDs are used to constrain $SO_2$ emissions by removing the second additive term on the right side of Eq. (1). Consequently, $\gamma$ is set

to unity. If the OMPS $SO_2$ tropospheric VCD error is set to 0.2 DU (Yang et al., 2013) for every pixel, the $SO_2$

observational term in the cost function (first term on the right side of Eq. (1)) over the North China Plain is much larger than that over Southwestern China (Fig. 2b), which yield high possibility to over-constrain the former and under-constrain the latter. The spatially unbalanced cost function is caused by cloud screening, as the number of observations over Southwestern China is much less than that over the North China Plain (Fig. 2a). To balance the cost function by accounting for this difference in the number of observations, $SO_2$ observation error is set to 0.2

DU multiplied by the square root of the number of OMPS overpasses that have $SO_2$ observation in the 2°x2.5° GEOS-Chem grid cell. The balance approach essentially normalizes observation terms in the cost function by the observation counts, which has been used in our study (Xu et al., 2015) to optimally invert aerosol optical properties from the skylight polarization and intensity measurements by AErosol RObotic NETwork (AERONET).

In the second experiment, E-$NO_2$, OMPS $NO_2$ tropospheric VCDs alone are used to constrain $NO_x$ emissions by removing the first additive term on the right side of Eq. (1). Due to cloud screening, much more OMPS $NO_2$ observations exist over the North China Plain than over Southwestern China, which also could lead to a spatially unbalanced cost function if the OMPS $NO_2$ observation error is uniform. The OMPS $NO_2$ observation error is, however, assumed to be 0.011 DU (Yang et al., 2014) for every pixel in this study, regardless of location, because

the $NO_x$ emissions adjustments during the inverse modeling process are supposed to be mainly over the North China Plain where prior $NO_x$ emissions are much larger than those over Southwestern China. In this study, we optimize emission scale factors rather than the emissions themselves. As a result, emissions are adjusted mainly at locations where prior emissions are large and kept as zero for those (2°×2.5°) grid boxes of zero prior emissions.

In the third experiment, E-joint, both the $SO_2$ and $NO_2$ from OMPS are used simultaneously for two reasons. Firstly, $SO_2$ and $NO_2$ concentrations can affect each other through several pathways. For example, Qu et. al (2019b) showed that the change of $SO_2$ or $NO_x$ emissions lead to the changes of $O_3$ and OH concentrations, hence the changes of $SO_2$ and $NO_2$ oxidations. Here, we will explore how the optimization results may dependent on the uncertainty of ammonia emissions (as elaborated in Sect. 3.2.2). Secondly, the computational time is reduced by

~50% in the joint assimilation as compared to separate assimilations when computational resource are restricted to running individual inversions sequentially (as opposed to in parallel), and energy usage is also saved; the latter require the realization of GEOS-chem adjoint twice, while only once is needed by the former.

In the E-joint experiment, observational terms for $SO_2$ and $NO_2$ in the cost function should be balanced through setting $\gamma$ in Eq. (1). When it is not balanced, $SO_2$ observations have very little impact on the inversion results as the optimization algorithm will firstly minimize the observational term for $NO_2$ unless many more iterations than is computationally feasible are performed, which is caused by the fact that observational error and valid number of $NO_2$ observation are respectively smaller and larger than the counterparts of $SO_2$. We thus subjectively derive $\gamma$ in a non-arbitrary way in order to focus equally on both species, thereby tackling the imbalance in their observational constraints. In this manner, the cost function is defined to serve the purpose of joint inversion of $SO_2$ and $NO_2$ emissions. Initially, we set $\gamma$ to be the ratio of number of $NO_2$ observations to the number of $SO_2$ observations. This approach is not feasible here as the $SO_2$ observational error in E-$SO_2$ is much larger than the $NO_2$ observational error in E-$NO_2$; not only does the number of observations play a role, but the observation error also has important impacts on balancing the cost function. If $\gamma$ is simply set as unity, the $NO_2$ observational term in Eq. (1) is a factor of ~200 larger than the $SO_2$ observational term, which can lead to OMPS $SO_2$ in the E-joint experiment to be negligible. Consequently, to balance the two terms, $\gamma$ is set as 200 (ratio of observational term in E-$NO_2$ to that in E-$SO_2$) in E-joint and sensitivity experiments of using different values of $\gamma$ are conducted (see section 3.2.2). Similar balance approach that adjusts contribution of observation terms in the cost function is used in the past work that assimilates satellite trace gas retrievals to invert emissions (Qu et al., 2019b) or invert the aerosol optical properties from skylight polarization measurements of AERONET (Xu et al., 2015).

### 3.2.2 Sensitivity experiments

To investigate the impacts of data quality control and spatially balancing the cost function on optimizing $SO_2$ emissions only, we design two sensitivity experiments. The first is E-$SO_2$-noQC-noBL that is similar to E-$SO_2$ except that: (1) OMPS $SO_2$ retrievals in the 2°x2.5° grid cell where the prior GEOS-Chem simulation is less than 0.1 DU are also assimilated, i.e. without QC; (2) OMPS $SO_2$ observation error is set as 0.2 DU for every pixel, which means we do not spatially balance the cost function. The second sensitivity experiment is E-$SO_2$-noBL in which the cost function is not spatially balanced, and it uses the same setting as E-$SO_2$ except for assuming an observation error of 0.2 DU uniformly.

To evaluate the effect of $\gamma$ (of 200) in E-joint, we further test $\gamma$ values of 20, 50, 100, 300, 500, 1000, 1500, and 2000 in the joint inversions; hereafter these experiments are named E-joint-d$\gamma$. Through these sensitivity experiments, we study the proper $\gamma$ range for jointly assimilating OMPS $SO_2$ and $NO_2$. In future studies that may be conducted to jointly assimilate OMPS $SO_2$ and $NO_2$ for other months to obtain a long-term optimized emission

inventory, it is proposed to set proper γ values for each month based on the range with easy adjustment according

to the numbers of OMPS $SO_2$ and $NO_2$ observations and their associated errors.

$NH_3$ emissions are not optimized in our inverse modeling and yet their uncertainty is up to 153% over China (Kurokawa et al., 2013). Thus, it is important to evaluate how this uncertainty may affect posterior $SO_2$ and $NO_x$ emissions. Wang et al. (2013) emphasized the importance of controlling $NH_3$ to alleviate $PM_{2.5}$ pollution over

China, however it could worsen acid rain (Liu et al., 2019). Changes of $NH_3$ emissions is expected to change ammonium and nitrate aerosol concentrations, or the aerosol surface area for heterogeneous $N_2O_5$ chemistry, hence affecting $NO_2$ concentrations or posterior $NO_x$ emissions in the inverse modeling. The change of posterior $NO_x$ emissions is expected to lead to the change of posterior $SO_2$ emissions in the joint inverse modeling. Thus, we shall investigate if $NH_3$ emissions are reduced to 50% and 20%, how the optimized $SO_2$ and $NO_2$ emission

inventories would change. Correspondingly, all these experiments are summarized in Table 2. E-$SO_2$-$0.5NH_3$, E-$NO_2$-$0.5NH_3$, and E-joint-$0.5NH_3$-γ500 in Table 2 are similar to E-$SO_2$, E-$NO_2$, and E-joint-dγ (γ=500) in Table 1, respectively, but $NH_3$ emissions are set to 50% of the original values. Similarly, E-$SO_2$-$0.2NH_3$, E-$NO_2$-$0.2NH_3$, and E-joint-$0.2NH_3$- γ500 are the scenarios that $NH_3$ emissions are set to 20% of the original values.

### 3.3 Evaluation statistics

We use linear correlation coefficient (R), root mean square error (RMSE), mean bias (MB), normalized mean bias (NMB), normalized standard deviation (NSD), and normalized centered root mean square error (NCRMSE) as measures to evaluate GEOS-Chem $SO_2$ and $NO_2$ VCD simulations with satellite (OMPS and OMI) observations. NSD is the ratio of the standard deviation of the simulation to the standard deviation of the observation. NCRMSE is similar to RMSE, but the impact of bias is removed. This is shown in Eq. (2), where $i$ is the $i$th grid cell, $N$ is

the total number of grid cells, $M_i$ and $O_i$ are the $i$th GEOS-Chem simulation and satellite observation, respectively, and $\bar{M}$ and $\bar{O}$ are averages of GEOS-Chem simulation and satellite observation, respectively. A composite summary of these statistics is provided by the Taylor diagram (Taylor, 2001) which is a quadrant which summarizes R (shown as cosine of polar angle), NSD (shown as radius from the quadrant center), and NCRMSE (shown as radius from expected, satellite observation, point, which is located at the point where R and NSD are

unity).

$$NCRMSE = \frac{\sqrt{\frac{1}{N}\sum_{i=1}^{N}[(M_i - \bar{M}) - (O_i - \bar{O})]^2}}{\sqrt{\frac{1}{N}\sum_{i=1}^{N}(O_i - \bar{O})^2}} \quad (2)$$

## 4. Results

### 4.1 Separate and joint assimilations of SO$_2$ and NO$_2$

### 4.1.1 Self-consistency check

The cost functions are reduced by 41.6%, 27.6%, and 28.6% for E-SO$_2$, E-NO$_2$, and E-joint, respectively, and the results are shown in Fig. 3. Noticeably, hot spots of SO$_2$ VCDs over the North China Plain and the Sichuan Basin are shown in the OMPS observations (Fig. 3a), prior (Fig. 3b), posterior E-SO$_2$ (Fig. 3c), and posterior E-joint (Fig. 3d) simulations, however the prior simulation has an NMB of 106.5% (Fig. 3i) when compared with OMPS. The SO$_2$ NMB (106.5%) between GOES-Chem prior simulation and OMPS is much larger than the NMB (-6.8%,

Fig S1) caused by the difference of SO$_2$ vertical profiles between OMPS SO$_2$ retrieval algorithm and current prior simulation; thus averaging kernel is not considered in the OMPS SO$_2$ observation operator. This large positive NMB decreases to 13.0% and 38.3% in the posterior E-SO$_2$ (Fig. 3j) and E-joint (Fig. 3k) simulations with an RMSE decreasing from 0.42 DU to 0.13 DU and 0.20 DU and R increasing from 0.62 to 0.72 and 0.64, respectively. Large NO$_2$ values are found over the North China Plain and Eastern China with large NO$_x$ emissions

from the transportation sector (Fig. 3e-h). Comparing with OMPS NO$_2$, GEOS-Chem results have an RMSE of 0.05 DU in the prior simulation (Fig. 3l) and reduce to 0.02 DU and 0.03 DU for E-NO$_2$ (Fig. 3m) and E-joint (Fig. 3n), with R increasing from 0.95 to 0.99 and 0.98, respectively.

Similarly, the averaging kernel is not considered in the OMPS NO$_2$ observation operator for optimization for the

following reasons. First, the OMPS NO$_2$ retrieval differences due to the profile differences can lead to a NMB of -7.5% (Fig S2), which is still smaller than the prior GEOS-Chem simulation NMB (10.9%, Fig. 3l). Second, a NMB of 10.9% for model NO$_2$ VCD simulation is not a very large value, as the difference between satellite NO$_2$ VCD retrievals and ground-based measurements could be comparable to this value. For example, Krotkov et al. (2017) shows that OMI NO$_2$ VCD retrievals, on average, are ~10% larger than ground-based FTIR spectrometer.

Thus, current research should mainly focus on the change of the spatial distribution (such as linear correlation coefficient) rather than bias of prior and posterior GEOS-Chem NO$_2$ VCD simulation. Finally, given that linear correlation coefficient between OMPS retrievals and that are modified through integration of averaging kernel and NO$_2$ vertical profile from this study is as large as 0.99, averaging kernel is not treated in the OMPS NO$_2$ observation operator. In general, the E-SO$_2$ and E-NO$_2$ posterior simulations show better results than E-joint,

which may be affected by the value of $\gamma$, which we will discuss in Sect. 4.3.

### 4.1.2 Emissions

The anthropogenic $SO_2$ and $NO_x$ prior MIX emissions for October 2010 and posterior emissions from E-$SO_2$, E-$NO_2$, and E-joint for October 2013 are shown in Fig. 4. $SO_2$ and $NO_x$ hot spots are found in the prior emissions over both the North China Plain and Eastern China, while large $SO_2$ emissions are also at Southwestern China.

Anthropogenic $SO_2$ emissions over China are 1166 Gg S in prior MIX for October 2010 (Fig. 4a), dropping 418 Gg S (Fig. 4b) and 306 Gg S (Fig. 4c), or 35.8% and 26.2%, in E-$SO_2$ and E-joint, respectively, for October 2013. The differences between the estimates of this study and the MIX emission inventory, however, should not be considered as trends, and they are derived from different approaches. Posterior E-joint total anthropogenic $SO_2$ emissions are 112 Gg, or 15% larger than E-$SO_2$ over China (Fig. 4e). Regionally, positive differences between

E-joint and E-$SO_2$ anthropogenic $SO_2$ emissions are over most areas of Central China and Eastern China, and relative difference is up to 100% over Shanxi province. (Fig. 4f). Girds with large differences are generally in locations where prior anthropogenic $SO_2$ emissions are larger, which means the pattern is affected by the fact the algorithm optimize emission scale factors rather than emissions directly. Anthropogenic $NO_x$ emissions over China  are reduced by 5.8% and 6.5% , from 714 Gg N in prior MIX for October 2010 (Fig 4g) to 672 Gg N (Fig.

4h) in E-$NO_2$ and 667 Gg N (Fig. 4i) in E-joint for October 2013. Although the relative difference between E-joint and E-$NO_2$ proved to be less than 2% in terms of total anthropogenic $NO_x$ emissions over China (Fig. 4k), it is up to 40% over Shanxi province, and both grids with large positive differences and grids with large negative differences exist over North China Plain (Fig. 4 l).

### 4.1.3 Independent evaluation with OMI data

The optimized emission inventories are evaluated by comparing prior and posterior GEOS-Chem simulations of $SO_2$ and $NO_2$ with OMI VCDs as shown in Fig. 5. We only focus on regions covered by OMPS observations, although smaller changes of emissions exist in outskirt regions where OMPS observations are not used. High $SO_2$ levels are shown over the North China Plain and the Sichuan Basin in both the prior and posterior simulations while OMI only observes hot spots over the former region (Fig. 5a-d). When validating with OMI $SO_2$ VCDs, the

NMB is ~300% in the prior simulation, and it reduces to ~100% in E-$SO_2$ and ~130% in E-joint (Fig. 5i). Not only is the NMB reduced, but the spatial distributions are also improved with the NCRMSE reducing from ~1.6 in the prior simulation to ~0.7 in E-$SO_2$ and ~0.8 in E-joint, which is much closer to ~0.6 when comparing OMPS observations with OMI observations (Fig. 5i). For $NO_2$, OMI observations and the prior and posterior simulations show large $NO_2$ concentrations over the North China Plain and Eastern China (Fig. 5e-h). The improvements for

E-NO$_2$ and E-joint are reflected in terms of R when evaluating with OMI tropospheric VCDs, although the two experiments show larger negative NMB than the prior simulation (Fig. 5j). In all the evaluations, OMI SO$_2$ and NO$_2$ VCD retrievals are not corrected by calculating new air mass factors that are derived from integrating scattering weights and corresponding vertical profiles of GEOS-Chem simulations of this study. However, Fig. S4 shows similar improvements if new air mass factors are applied, although statistic metric values are slightly 420 different.

Here, OMPS observations and GEOS-Chem simulations are compared with OMI observations as an evaluation of posterior emission inventories, but it is not assumed that OMI provides the true status of SO$_2$ and NO$_2$ in the atmosphere. OMI and OMPS observe the same trend direction of SO$_2$ (NO$_2$) over China, but the strengths of trend 425 are quite different (Wang and Wang, 2020). OMPS SO$_2$ average is ~0.14 DU, or ~95% larger than OMI SO$_2$, and the R of the two products is 0.81 (Fig. 6b). Thus, it is reasonable that posterior SO$_2$ is larger than OMI observations by ~100% in E-SO$_2$ and ~130% in E-joint. OMPS NO$_2$ is ~24% smaller than OMI (Fig. 6d), which explains why the posterior NO$_2$ simulations have larger negative NMB than the prior simulation when compared with the OMI observations. Our analysis also shows that the systematic difference among various satellite products for the same 430 species (such as SO$_2$ or NO$_2$) can lead to biases in constraining emissions, but the posterior GEOS-Chem simulations still show in terms of the spatial distribution of SO$_2$ and NO$_2$.

**4.2 The impacts of QC and spatial balance**

The results of E-SO$_2$-noQC-noBL and E-SO$_2$-noBL are compared with E-SO$_2$ to show the impacts of QC and spatial balance. Both OMPS retrievals and the GEOS-Chem prior simulations show that SO$_2$ VCDs over Inner 435 Mongolia and the Sichuan Basin (grid cells M and S, respectively in Fig. 7) are smaller than those over the North China Plain; this pattern reverses in the posterior E-SO$_2$-noQC-noBL simulation where SO$_2$ over the North China Plain becomes smaller than that over grid cells M and S. Grid cell M becomes more reasonable after conducting the data quality control by removing OMPS SO$_2$ in any grid cells where prior GEOS-Chem SO$_2$ VCDs are less than 0.1 DU (e.g., as in E-SO$_2$-noBL, as shown in Fig. 7d). QC helps to improve models over grid cell M, as the 440 data removed are close to Inner Mongolia, and are generally less than 0.1 DU, which are comparable to the retrieval error. SO$_2$ over grid cell S from E-SO$_2$-noBL (Fig. 7d) is, however, still larger than that over the North China Plain, compared with the better spatial pattern from E-SO$_2$ (Fig. 3c). Thus, QC and spatial balancing of the cost function together improve the spatial pattern of the posterior GEOS-Chem SO$_2$ VCD simulation.

**4.3 The impacts of γ on joint assimilations**

In addition to setting γ as 200 in E-joint, we test the impacts of using various γ values on joint assimilation in E-joint-dγ for October 2013. All the $SO_2$ and $NO_2$ VCDs from prior and posterior E-joint and E-joint-dγ simulations are compared with OMPS counterparts (Fig. 8a-b). Regardless of the γ values used, all the posterior simulations of $SO_2$ show smaller NMB and NCRMSE than the prior simulation when validating against OMPS and OMI counterparts, but the extents of improvement vary. When γ is 20, 50, or 100, the $SO_2$ terms are obviously under-

constrained, and GEOS-Chem $SO_2$ NCRMSE, evaluated with OMPS observations, changes from ~1.8 in the prior simulation to in the range of ~1.4 to ~1.7 in the posterior E-joint-dγ simulations, which are much larger than ~0.7 in E-$SO_2$ (Fig. 8a). Similarly, when γ is no larger than 100, the bias of GEOS-Chem $SO_2$, validated with OMPS observations, only reduces from ~100% to ~75%, compared to ~25% in E-$SO_2$ (Fig. 8a), and the posterior $SO_2$ emissions are in the range of 1055 Gg S to 1143 Gg S, which are much larger than 748 Gg S from E-$SO_2$ (Table

3). When γ is in the range of 200 to 2000, the $SO_2$ simulation results and emissions from joint assimilations are more similar to that from E-$SO_2$ than that with γ no larger than 100 (Fig. 8a and Table 3). Similar to $SO_2$, the GEOS-Chem simulations of $NO_2$ in the sensitivity experiments improve in terms of R and NCRMSE in all joint assimilation tests, but the significance of γ is less than that for $SO_2$. $NO_2$ NCRMSE is ~0.4 in the prior simulation when evaluating with OMPS counterparts, compared to the range of ~0.2 to ~0.25 in E-joint, E-joint-dγ and E-

$NO_2$ (Fig. 8b). The posterior $NO_x$ emissions are in the range of 662 Gg N to 682 Gg N, compared with 672 Gg N in E-$NO_2$ (Table 3).

    The impacts of γ are also reflected when evaluating $SO_2$ and $NO_2$ simulations with OMI retrievals (Fig. 8c-d). Small γ values of 20, 50, and 100 lead to a much larger bias and NCRMES for $SO_2$ from E-joint-dγ than that from

E-$SO_2$. For $NO_2$, these small γ values make results from E-joint-dγ very similar to that from E-$NO_2$.

    Considering all of the above analyses, the results with γ in the range of 200 to 2000 are deemed acceptable. The E-joint-dγ (200≤ γ ≤2000) emissions are within -3% to 15% of E-$SO_2$ for $SO_2$ and ±2% of E-$NO_2$ for $NO_x$ in terms of total anthropogenic $SO_2$ and $NO_x$ emissions over China. When evaluating with OMPS observations, the

NCRMSE of using the posterior emissions from the separate and joint (200≤ γ ≤2000) inversions are ~60% and ~45%-60% smaller than that of using the prior emissions for $SO_2$, respectively, and ~50% and ~38%-50% smaller than that of using the prior emissions for $NO_2$, respectively.

When evaluating with OMI retrievals, joint inversion shows better results than separate inversion for $SO_2$ or $NO_2$, but not both, depending on the value of $\gamma$. When $\gamma$ is 20, 50, or 100, $NO_2$ NCRSME for E-joint-d$\gamma$ improvement appears to be smaller than that for E-$NO_2$, but $SO_2$ NCRSME for E-joint-d$\gamma$ is larger than that for E-$SO_2$. Conversely, when $\gamma$ is 1000, 1500, or 2000, $SO_2$ NCRSME for E-joint-d$\gamma$ is smaller than that for E-$SO_2$, but $NO_2$ NCRSME for E-joint-d$\gamma$ is larger than that for E-$NO_2$. This is similar to the findings by Qu et al., (2019b) in which the months when joint inversion shows better result than separate inversion for $SO_2$ ($NO_2$) have worse result for $NO_2$ ($SO_2$). The benefit of joint inversion for improving only one species is similar to Qu et al. (2019b) and is likely due to the complicated relationship between these two species through different chemical pathways. For example, $O_3$ and OH are key species that connect the chemistry of $SO_2$ and $NO_2$ and aerosols can affect the photolysis and heterogenous chemistry. Hence, while joint inversion to improve both species cannot be demonstrated here, it should be reviewed as the first step of simultaneously assimilating multiple species (including AOD, $NH_3$, and other trace gases) to optimize emissions. Until then, the system is not ready to holistically evaluate the benefits of joint assimilation to improve the model in a systematic manner. It is worthy noting that Xu et al. (2013) showed the feasibility of using MODIS cloud-free radiance to optimize emissions of $SO_2$ and $NO_2$ at the same time. Future research should add the aerosol optical depth or visible reflectance (as well as tropospheric $O_3$ if reliable) as constraints to further evaluate the benefits of joint assimilation for improving model overall performance in a systematic matter.

### 4.4 The impacts of $NH_3$ emission

In the single-species inversions, $NH_3$ emission uncertainty has weaker impacts on posterior $SO_2$ emissions than $NO_x$ emissions. Posterior $SO_2$ emissions over China are 748 Gg S in the 100% $NH_3$ emission scenario (E-$SO_2$), and they only slightly reduce to 747 Gg S and 745 Gg S when $NH_3$ emissions are 50% (E-$SO_2$-0.5$NH_3$) and 20% (E-$SO_2$-0.2$NH_3$) of the original values, respectively (Table 4). The largest relative changes at model-grid-cell scale are only -2.5% (Fig. 9a) for E-$SO_2$-0.5$NH_3$ for and -4.7% (Fig. 9b) for E-$SO_2$-0.2$NH_3$. All these results can be explained by considering how changes of $NH_3$ can potentially impact the lifetimes of $SO_2$ and $NO_2$ and hence affect $SO_2$ and $NO_2$ VCD simulations. When the $NH_3$ emissions decrease to 50%, and 20% $SO_2$ VCDs only increase up to 3.8% and 6.1%, respectively, in some grid cells over the Sichuan Basin in the prior simulations, and these changes are even much smaller over the North China Plain (Fig. 10a-b), as $NH_3$ has no direct effect on the life cycle of $SO_2$. This is understandable because in GEOS-Chem, once $SO_2$ is oxidized to $H_2SO_4$, $SO_4^{2-}$ remains as particulate sulfate regardless it is neutralized by $NH_3$ or not (Wang et al., 2008). Hence, the reductions

of NH$_3$ to 50% and 20% overall has minimal (negligible) impact on SO$_2$ amount in the prior simulation and the posterior separate SO$_2$ emission inversion.


Although the posterior NO$_x$ emissions in the scenarios of 50% (E-NO2-0.5NH3) and 20% (E-NO2-0.2NH3) NH$_3$ emission experiments of the original values are 5 Gg N (0.7%) and 19 Gg N (2.8%), respectively, which are smaller than those when using the original (E-NO2) NH$_3$ emissions over China (Table 4), the reduction is up to -4.0% (Fig. 9e) for E-NO2-0.5NH3 and -9.1% (Fig. 9f) for E-NO2-0.2NH3 in individual grid cells. These decreases

are understood by simultaneous reduction of nitrate by 59.5% (Fig. 12h vs. 12g) and 80.5% (Fig 12i vs. 12g) and ammonium by 39.6% (Fig. 12n vs. 12m) and 67.5% (Fig. 12o vs. 12m), which leads to large reduction of the hydrated aerosol surface area for heterogeneous N$_2$O$_5$ chemistry at night, hence overall NO$_2$ lifetime (Fig. 10c-d). N$_2$O$_5$ normally forms at night by reaction between NO$_2$ and NO$_3$, and thermally decomposes back to NO$_2$ and NO$_3$ (Seinfeld and Pandis, 2016), and hence the amount of N$_2$O$_5$, NO$_2$, and NO$_3$ are in equilibrium through the

reversible reaction. Since the hydrolysis of N$_2$O$_5$ to form HNO$_3$ mainly occurs on hydrated aerosol particles (Seinfeld and Pandis, 2016), the decrease of hydrated aerosol surface area (due to reduction of NH$_3$ emission) leads to less hydrolysis of N$_2$O$_5$ (an important sink for atmospheric NO$_x$) and subsequently more NO$_2$ to be in the equilibrium with N$_2$O$_5$ at night. As a result, the reduction of NH$_3$ emissions further increases the positive bias in the prior NO$_2$ simulations when comparing with OMPS observations, and to compensate such large positive bias,

non-negligible decreases in the posterior NO$_x$ emissions are required (Fig. 9 e and f). The reduction of nitrate and ammonium aerosols can also increase sunlight reaching troposphere, hence photolysis O$_3$ and NO$_2$. Figure S5 separates the impacts of increase of photolysis O$_3$ and NO$_2$ and decrease heterogeneous N$_2$O$_5$ chemistry on NO$_2$ lifetime and shows that the former is negligible compared with the latter.

The decreases of posterior SO$_2$ and NO$_x$ emissions in the joint inversions caused by the reduction of NH$_3$ emissions are stronger than that in the separate inversions (Table 4 and Fig. 9). Although the changes of NH$_3$ emissions only have slight impacts on the SO$_2$ separate inversions (E-SO2, E-SO2-0.5NH3, and E-SO2-0.2NH3), the posterior SO$_2$ emission is 802 Gg S in E-joint-d$\gamma$ ($\gamma$=500), down to 783 Gg S (decreasing by 2.4%) and 746 Gg S (decreasing by 7.0%) in E-joint-0.5NH3- $\gamma$500 and E-joint-0.2NH3- $\gamma$500, respectively (Table 4); in some

grid cells, the relative reductions are up to -9.0% (Fig. 9c) for E-joint-0.5NH3- $\gamma$500 and -27.7% (Fig. 9d) for E-joint-0.2NH3- $\gamma$500. For posterior NO$_x$ emissions at the grid cells, the relative changes are up -15.2% (Fig. 9g) for E-joint-0.5NH3- $\gamma$500 and -29.4% (Fig. 9h) for E-joint-0.2NH3- $\gamma$500 with respect to E-joint-d$\gamma$ ($\gamma$=500).

**4.5 Aerosol responses to emission changes**

Although SO$_2$ emissions over the North China Plain (E-joint-d$\gamma$ ($\gamma$=500)) have decreased by more than 50%, and NO$_x$ emissions have also been reduced, reductions of Sulfate-Nitrate-Ammonium (SNA) Aerosol Optical Depth (AOD) over the same region are only up to 10% (Fig. 11). This is because the North China Plain is mainly polluted by nitrate rather than sulfate (Fig. 12a-l), and the reduction of SO$_2$ emissions will increase nitrate loadings in the atmosphere (Fig. 12g-l), which is also consistent with Kharol et al. (2013)'s research that shows nitrate concentrations decrease as SO$_2$ emissions increase; the reduction of SO$_2$ emissions lead to less H$_2$SO$_4$ to react with NH$_3$, which further favor the reaction of HNO$_3$ and NH$_3$ to form nitrate. As NH$_3$ emissions change reduce by 50% and 80% ammonium column loadings decrease by ~40% and ~70% (Fig. 12g-l), respectively, and nitrate column loadings decrease even by ~70% and ~90%, respectively (Fig. 12m-r).

**5. Discussion and conclusions**

We developed 4D-var observation operators for assimilating OMPS SO$_2$ and NO$_2$ VCDs to constrain SO$_2$ and NO$_x$ emissions through GEOS-Chem adjoint model. The approach is applied for case study over China for October 2013 at 2°x2.5° resolution and the MIX 2010 is used as the prior emission inventory. Several experiments of assimilating OMPS SO$_2$ and NO$_2$ separately and jointly are conducted, and SO$_2$ and NO$_2$ VCDs from the GEOS-Chem prior and posterior simulations are compared with counterparts from OMPS and OMI.

OMPS SO$_2$ and NO$_2$ retrievals are separately and jointly used to constrain their corresponding emissions. In the single-species inversions, posterior anthropogenic SO$_2$ and NO$_x$ emissions are 748 Gg S and 672 Gg N for October 2013, down from 1166 Gg S and 714 Gg N in the prior MIX for October 2010, respectively. In the joint inversions of assimilating OMPS SO$_2$ and NO$_2$ simultaneously, the cost function is balanced according to the values of observational terms rather than the number of observations. When the cost function is well balanced ($\gamma$ in the range of 200 to 2000), the results of the joint inversions are within -3% to 15% of the single-species inversion for total anthropogenic SO$_2$ emissions and ±2% for total anthropogenic NO$_x$ emissions. However, the differences between the separate and joint inversions are up to 100% and 40% in some model grid cells for anthropogenic SO$_2$ and NO$_x$ emissions, respectively. In comparison to OMPS observations, NCRMSE from joint inversions ($\gamma$ in the range of 200 to 2000) is reduced by ~45%-~60% for SO$_2$ and ~38%-~50% for NO$_2$, respectively, which is close to the ~60% reduction from the SO$_2$ inversion and the ~50% reduction from the separate NO$_2$ inversion. To obtain posterior emissions for both SO$_2$ and NO$_x$, the computational time for the joint inversion is only about ~50%

of the single-species inversions, when the latter are computed sequentially. Moreover, posterior GEOS-Chem $SO_2$ and $NO_2$ show improvements in terms of R when comparing against OMI observations, and the increase of posterior GEOS-Chem $NO_2$ negative NMB is ascribed to that the average of OMPS $NO_2$ over China is smaller than the OMI counterpart. Above all, the posterior emission increases the GEOS-Chem simulated spatial distributions of $SO_2$ and $NO_2$.

Both data quality control and spatially balancing the cost function play an important role for constraining $SO_2$ emissions. OMPS $SO_2$ retrievals over the regions where emissions are small are removed as VCDs are comparable to retrieval errors. A sensitivity study shows that if these data are included, it will lead to artifacts in the posterior $SO_2$ emission spatial distribution. Due to cloud screening, the number of OMPS $SO_2$ retrievals over the Sichuan Basin is much less than that over the North China Plain, which will lead to under-constraining over Sichuan Basin, if the observation error is assumed spatially constant. When the observation error is set based on the number of observations, the artifacts are avoided.

To investigate the impacts of the uncertainty of $NH_3$ emissions on posterior $SO_2$ and $NO_x$ emissions, several inverse modeling experiments are conducted by setting prior $NH_3$ emissions to as 50% and 20% of their original values. The reduction of $NH_3$ emissions can lead to a larger decrease of posterior $NO_x$ emissions and a smaller decrease of $SO_2$ emissions in separate assimilations, which ascribes to the $NO_2$ lifetime is more than the $SO_2$ affected by the change of $NH_3$ emissions. The impacts of $NH_3$ emissions uncertainty on both posterior $SO_2$ and $NO_x$ emissions in joint assimilations are stronger than separate assimilations.

Large $SO_2$ emissions are mainly produced over the Sichuan basin and the North China Plain, while AOD responses to the changes of $SO_2$ emissions are quite different over the two regions. The reduction in $SO_2$ emissions can effectively decrease AOD over the Sichuan Basin, while AOD declines only slightly over the North China Plain, which can be ascribed to (1) nitrate rather than sulfate is dominant over the North China Plain and (2) the reduction of $SO_2$ emissions facilitate the formation of additional nitrate. AOD over the North China Plain is mainly determined by $NO_x$ and $NH_3$ emissions rather than $SO_2$ emissions.

All emissions are constrained on the monthly scale and at the coarse spatial resolution of $2° \times 2.5°$ in this study, as OMPS observations are provided once per day at the resolutions as coarse as $50 \times 50$ $km^2$ at nadir and $50 \times 190$ $km^2$ at edge and the 4D-Var data assimilation at finer spatial resolution (on the order of 0.1 degree) would be

computationally prohibitive. The approach, however, has the potential for optimizing emissions at daily to weekly scale and in fine spatial  resolution (on order of ~ 10 km) from future satellite observations at high spatial and temporal resolutions. In particular, TEMPO (monitoring North America), GEMS (monitoring East Asia), and Sentinel-4 (monitoring Europe) are to be launched in the next several years, and all of these satellites will provide hourly $SO_2$ and $NO_2$ observations during the daytime with resolution of $2.1 \times 4.4$ $km_2$, $7 \times 8$ $km_2$, $8.9 \times 11.7$ $km_2$, respectively. Furthermore, in Part II of this work, we develop various downscale methods to apply these coarser-resolution top-down estimates of emissions for air quality forecasts and evaluate the forecasts with surface measurements, both at the finer spatial scale (Wang et al., 2019).

Author contributions. All authors designed the research; YW conducted the research; YW and JW wrote the paper; XX, DKH and ZQ contributed to writing.

Competing interests. The authors declare that they have no conflict of interest.

Acknowledgements. This research is supported by the National Aeronautics and Space Administration (NASA) through Aura program managed by Kenneth W. Jucks, ACMAP program (grant numbers: NNX17AF77G and 80NSSC19K0950) managed by Richard Eckman, and through TEMPO project as part of NASA's Earth Venture program (grant number SV7-87011 subcontracted from Harvard Smithsonian Observatory to the University of Iowa). We acknowledge the computational support from the High-Performance Computing group at The University of Iowa and Prof. Charles O. Stanier from The University of Iowa for insightful comments on the analysis of $SO_2$ and $NO_2$ lifetimes.

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

**Table 1. Different experimental design for using OMPS SO$_2$ and NO$_2$ to constrain corresponding emissions over China for October 2013.**

| Name | Data | SO$_2$ error[b] | NO$_2$ error | $\gamma$[c] | QC for SO$_2$[d] |
|---|---|---|---|---|---|
| E-SO$_2$ | SO$_2$ | 0.2 DU x $\sqrt{N}$ | NA | 1 | Yes |
| E-NO$_2$ | NO$_2$ | NA | 0.011 DU | NA | NA |
| E-joint | SO$_2$ and NO$_2$ | 0.2 DU x $\sqrt{N}$ | 0.011 DU | 200 | Yes |
| E-SO$_2$-noQC-noBL | SO$_2$ | 0.2 DU | NA | 1 | No |
| E-SO$_2$-noBL | SO$_2$ | 0.2 DU | NA | 1 | Yes |
| E-joint-d$\gamma$ | SO$_2$ and NO$_2$ | 0.2 DU x $\sqrt{N}$ | 0.011 DU | 20 to 2000[e] | Yes |

[a]See description of these names in detail in Set. 3.2.

[b]N in this column is number of OMPS overpass that have SO$_2$ observation in the 2x2.5 GEOS-Chem grid cell.

[c]$\gamma$ is a parameter used to balance SO$_2$ and NO$_2$ observation terms in the cost function.

[d]OMPS SO$_2$ retrievals in the 2x2.5 grid cell where the prior GEOS-Chem simulation is less than 0.1 DU are removed.

[e]All these $\gamma$ values (20, 50, 100, 300, 500, 1000, 1500, and 2000) are used.


**Table 2. Different experimental design for assessing the impacts of NH$_3$ emission inventories on using OMPS SO$_2$ and NO$_2$ to constrain corresponding emissions over China for October 2013[a].**

| Name[b] | Data | $\gamma$[c] | NH$_3$ emissions |
|---|---|---|---|
| E-SO$_2$-0.5NH$_3$ | SO$_2$ | NA | 50% |
| E-NO$_2$-0.5NH$_3$ | NO$_2$ | NA | 50% |
| E-joint-0.5NH$_3$- $\gamma$500 | SO$_2$ and NO$_2$ | 500 | 50% |
| E-SO$_2$-0.2NH$_3$ | SO$_2$ | NA | 20% |
| E-NO$_2$-0.2NH$_3$ | NO$_2$ | NA | 20% |
| E-joint-0.2NH$_3$- $\gamma$500 | SO$_2$ and NO$_2$ | 500 | 20% |

[a]Data quality control and observation errors are same as E-joint in Table 1.

[b]See description of these names in detail in Set. 3.2.

[c]$\gamma$ is a parameter used to balance SO$_2$ and NO$_2$ observation terms in the cost function.

**Table 3. Posterior anthropogenic emissions for October 2013 from E-joint, E-joint-d$\gamma$, E-SO$_2$ and E-NO$_2$.**

| Experiment name or $\gamma$ | 20 | 50 | 100 | 200 | 300 | 500 | 1000 | 1500 | 2000 | E-SO$_2$ or E-NO$_2$ |
|---|---|---|---|---|---|---|---|---|---|---|
| SO$_2$ [Gg S] | 1143 | 1110 | 1055 | 860 | 795 | 802 | 733 | 730 | 728 | 748 |
| NO$_x$ [Gg N] | 681 | 682 | 682 | 667 | 662 | 664 | 668 | 666 | 674 | 672 |

**Table 4. Posterior anthropogenic emissions for October 2013 under different NH$_3$ emission scenarios**

| Name | SO$_2$ emissions [Gg S] | NO$_x$ emission [Gg N] |
|---|---|---|
| E-SO$_2$ | 748 | NA |
| E-SO$_2$-0.5NH$_3$ | 747 | NA |
| E-SO$_2$-0.2NH$_3$ | 745 | NA |
| E-NO$_2$ | NA | 672 |
| E-NO$_2$-0.5NH$_3$ | NA | 667 |
| E-NO$_2$-0.2NH$_3$ | NA | 653 |
| E-joint-d$\gamma$ ($\gamma$=500) | 802 | 664 |
| E-joint-0.5NH$_3$- $\gamma$500 | 783 | 646 |
| E-joint-0.2NH$_3$- $\gamma$500 | 746 | 629 |

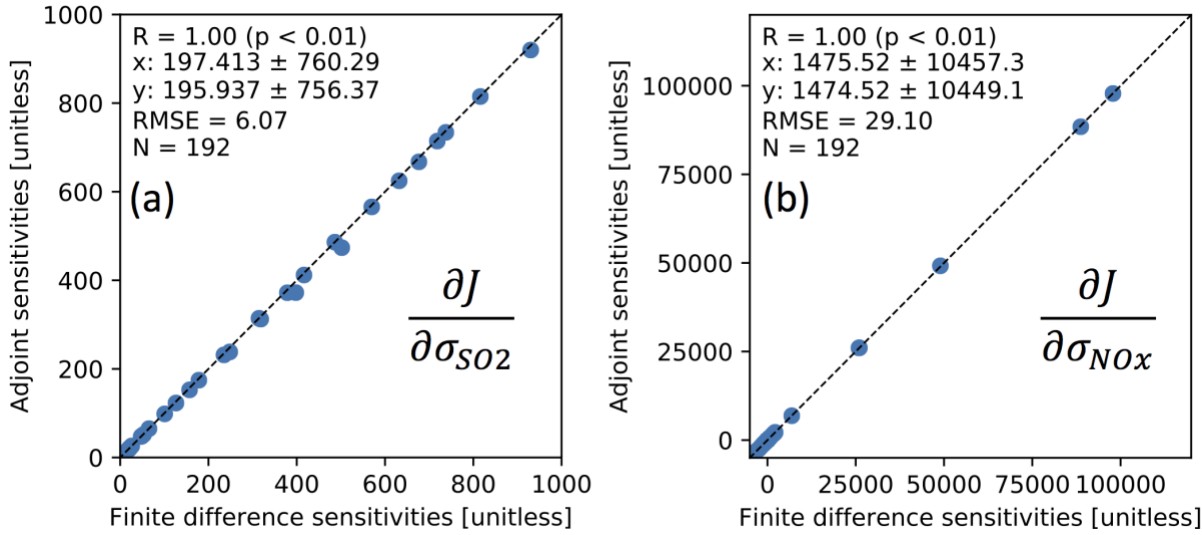

**Figure 1. Validation of adjoint model sensitivity through comparison to centered finite difference results for a 3-day simulation.** Shown here are the sensitivity of column cost function (penalty term is not included, and horizontal transport is turned off) with respect to logarithm of anthropogenic $SO_2$ (a) and $NO_x$ (b) emission scale factors: the 1:1 line (dotted), the number of grid columns (N), Root Mean Squared Error (RMSE), and correlation coefficient (R), and Means and standard deviations of finite difference sensitivity and adjoint sensitivity (x and y).


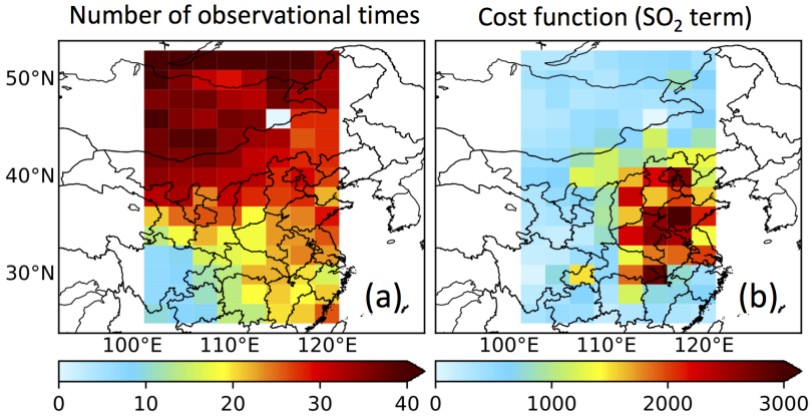


**Figure 2. (a) and (b) are the numbers of the OMPS overpass time that provides SO₂ VCD retrievals and SO₂ term in cost function at first iteration, respectively, in October 2013**

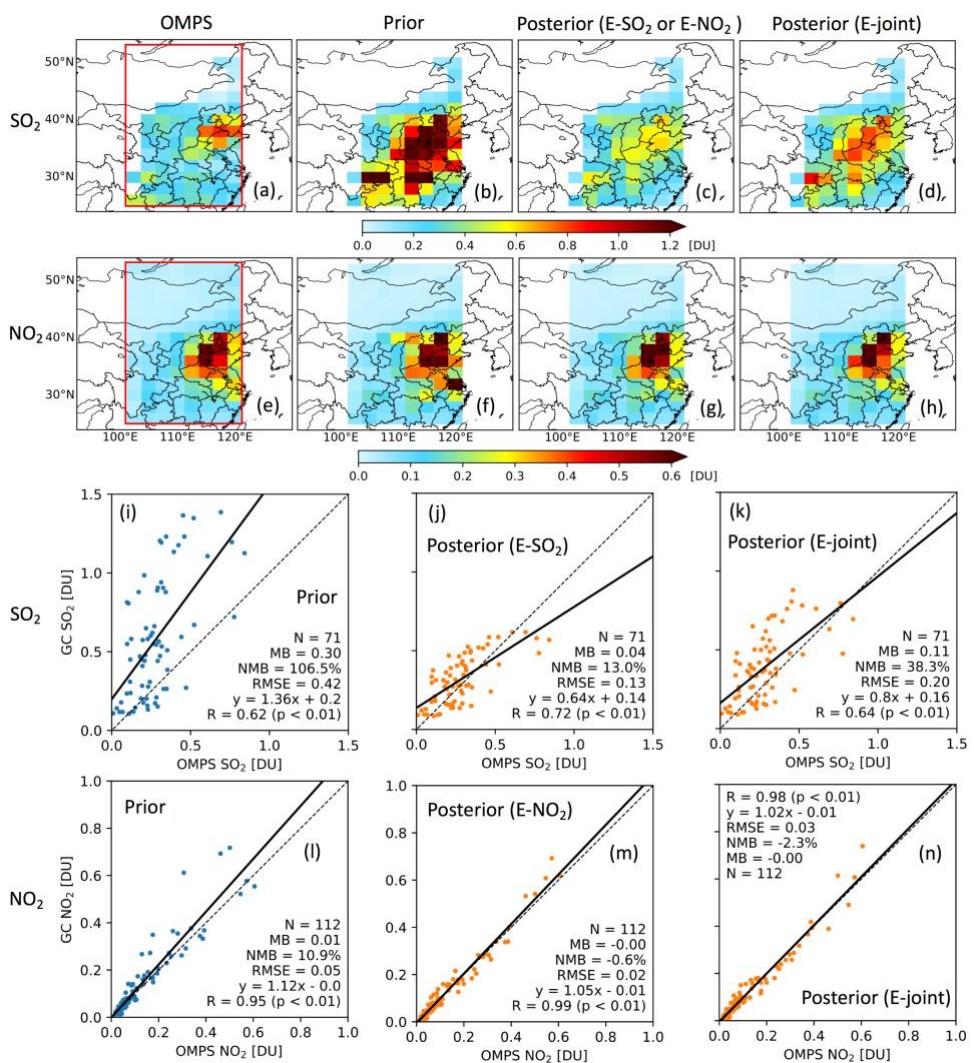

**Figure 3. Comparisons of VCDs of SO₂ and NO₂ from the OMPS and the GEOS-Chem prior and posterior simulations in October 2013 over China.** The first row is SO₂ VCDs from the OMPS (a), the prior simulation (b), the E-SO₂ posterior simulation (c), and the E-joint posterior simulation (d). The second row is NO₂ tropospheric VCDs from the OMPS (e), the prior simulation (f), the E-NO₂ posterior simulation (g), and the E-joint posterior simulation (h). The third row is the SO₂ VCD scatter plots of the GEOS-Chem prior (i), the E-SO₂ posterior (j), and the E-joint posterior (k) versus the OMPS, respectively. The last row is the NO₂ tropospheric VCD scatter plots of the GEOS-Chem prior (l), the E-NO₂ posterior (m), and the E-joint posterior (n) versus the OMPS, respectively. Linear correlation coefficient (R), linear regression equation, root mean squared error (RMSE), normalized mean bias (NMB), mean bias (MB), and number of observations (N) are shown over scatter plots.

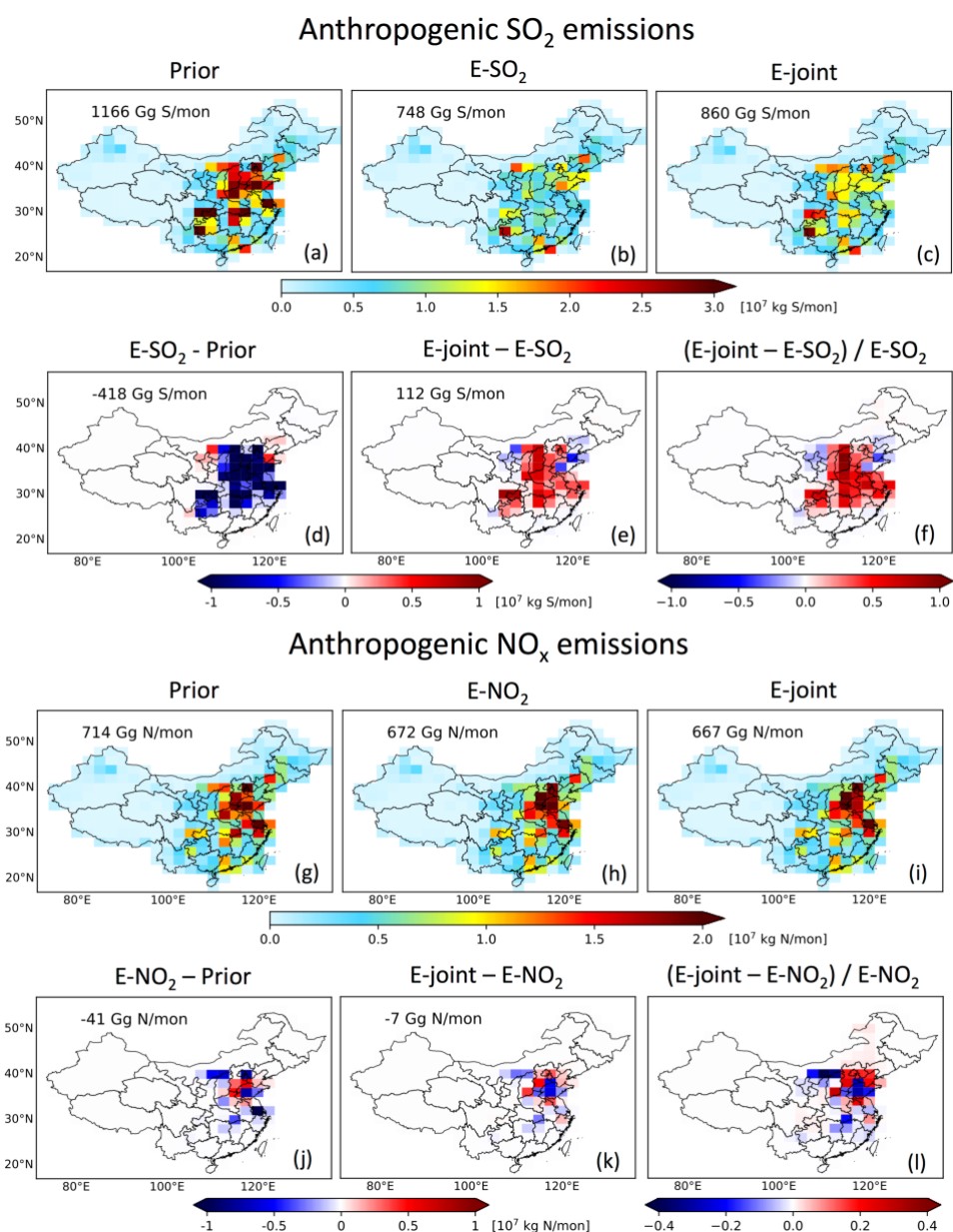

**Figure 4. The top is anthropogenic SO₂ emissions from prior MIX 2010 (a), posterior E-SO₂ (b), posterior E-joint (c), the difference between posterior E-SO₂ and prior MIX 2010 (d), the difference between posterior E-joint and posterior E-SO₂ (e), and the relative difference between posterior E-joint and posterior E-SO₂ (f) for October 2013. The bottom is similar to the top except that (1) it is for NOₓ and (2) E-SO₂ is replaced by E-NO₂.**


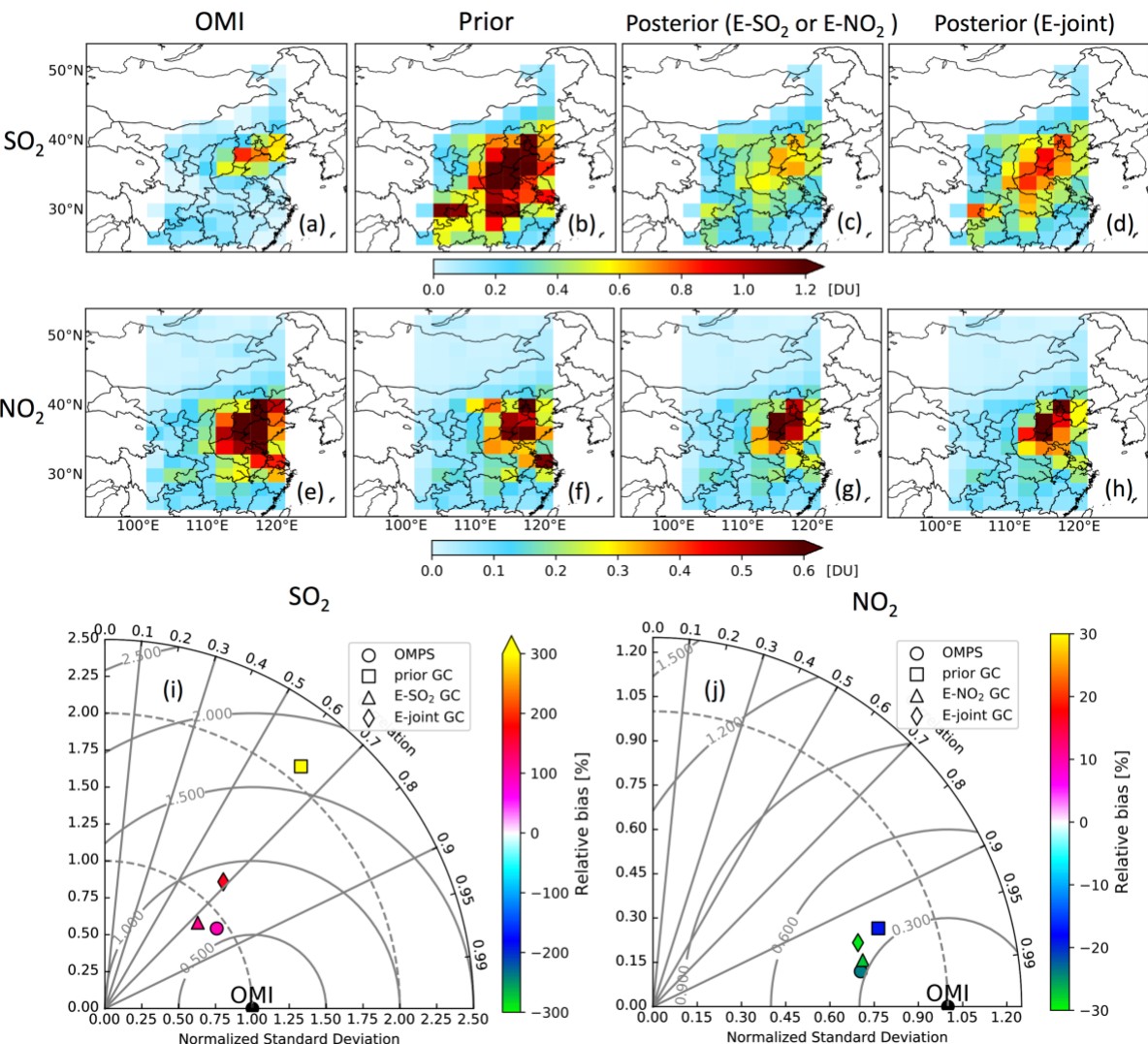

**Figure 5. Comparisons of VCDs of SO₂ and NO₂ from the OMPS and the GEOS-Chem prior and posterior simulations with that from the OMI in October 2013 over China. The first row is SO₂ VCDs from the OMI (a), the prior simulation (b), the E-SO₂ posterior simulation (c), and the E-joint posterior simulation (d). The second row is NO₂ tropospheric VCDs from the OMI (e), the prior simulation (f), the E-NO₂ posterior simulation (g), and the E-joint posterior simulation (h). The third row is Taylor diagrams for comparing GEOS-Chem simulations (squares for prior, triangles for posterior E-SO2 or E-NO2, and diamonds for E-joint) and OMPS observations (circles) with OMI SO2 (i) and NO2 (j).**

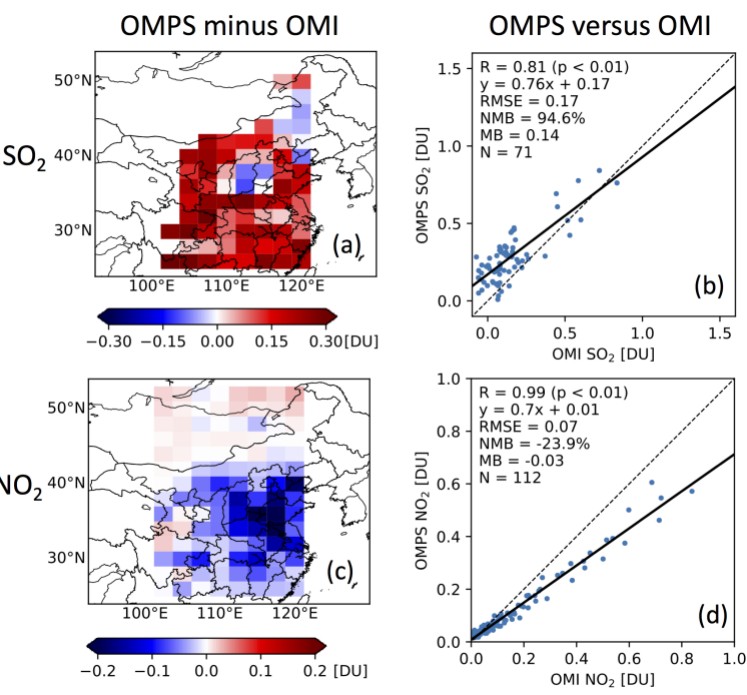

**Figure 6. (a) and (b) are the difference between OMPS and OMI SO₂ and scatter plot of OMPS versus OMI SO₂. (c) and (d) are similar (a) and (b), but for NO₂. Linear correlation coefficient (R), linear regression equation, root mean squared error (RMSE), normalized mean bias (NMB), mean bias (MB), and number of observations (N) are shown over scatter plots.**

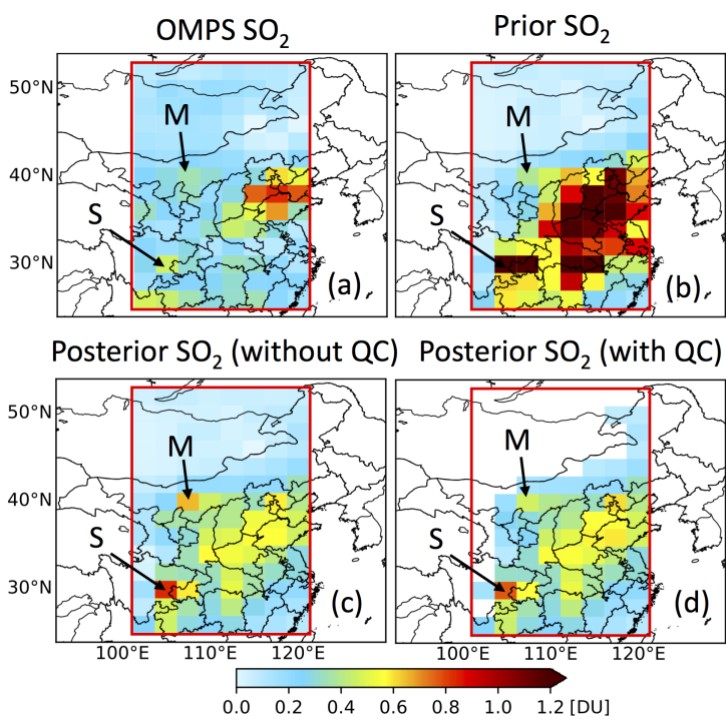

**Figure 7. SO₂ VCD in October 2013 from OMPS (a), prior GEOS-Chem simulation (b), posterior GEOS-chem simulation through using all OMPS data in the red box (c), and posterior GEOS-chem simulation through using only OMPS data that are in the grid cell where GEOS-Chem prior simulation of VCD is larger than 0.1 DU. For posterior simulation, we only plot SO₂ VCD over grid cells where OMPS data are used to constrain emissions. M and S point to a grid cell in Inner Mongolia and Sichuan basin, respectively.**



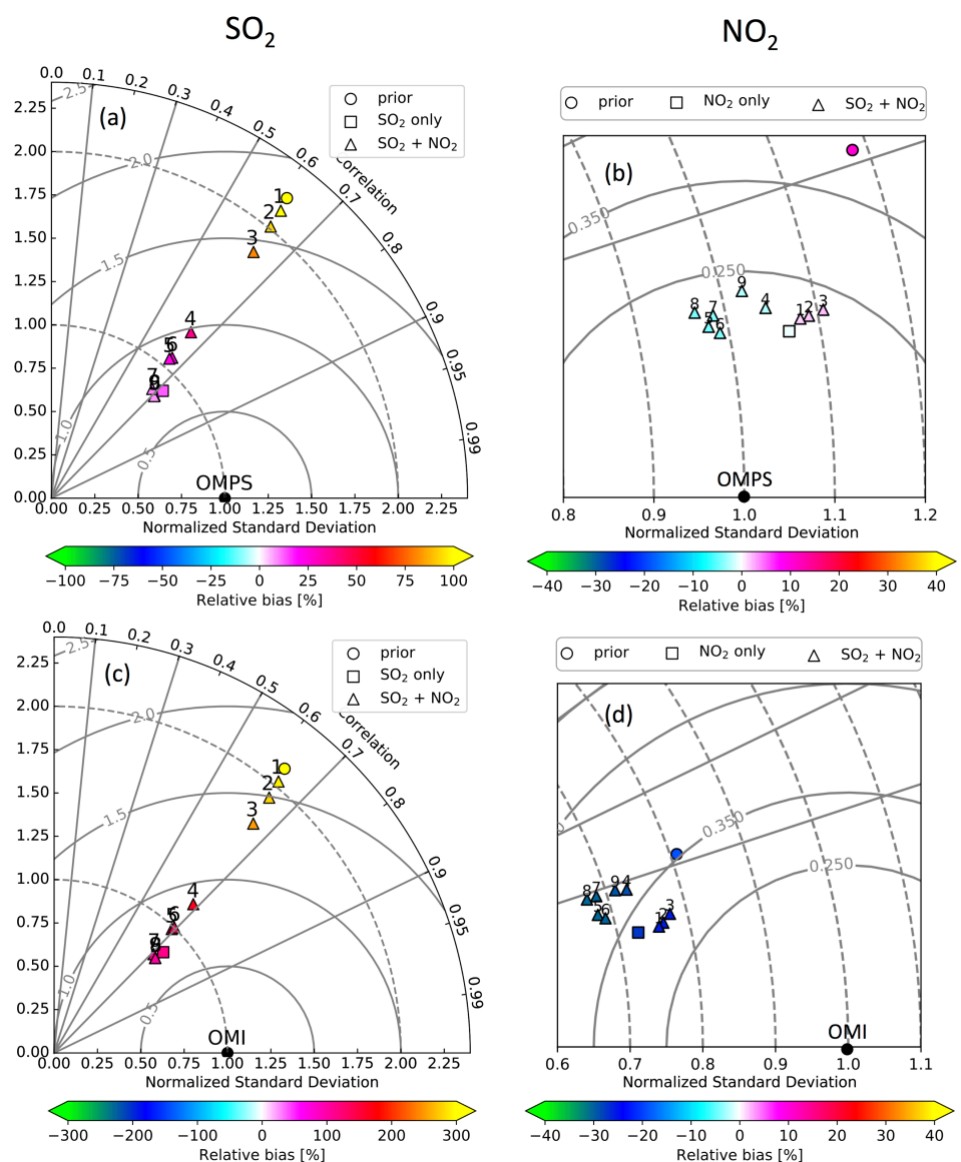

**Figure 8. Taylor diagram of comparing GEOS-Chem simulation with OMPS (a for SO$_2$ and b for NO$_2$) or OMI (c for SO$_2$ and d for NO$_2$) in October 2013. Circles, squares, and triangles represent GEOS-Chem simulations using prior MIX 2010 emissions, posterior emissions constrained by single species (E_SO2 for a and c, E_NO2 for b and d), and posterior emissions constrained through joint inversion (E_joint), respectively. Different triangles labeled by numbers represent different γ values in Eq. (1), and 1 through 9 correspond to 20, 50, 100, 200, 300, 500, 1000, 1500, and 2000, respectively.**

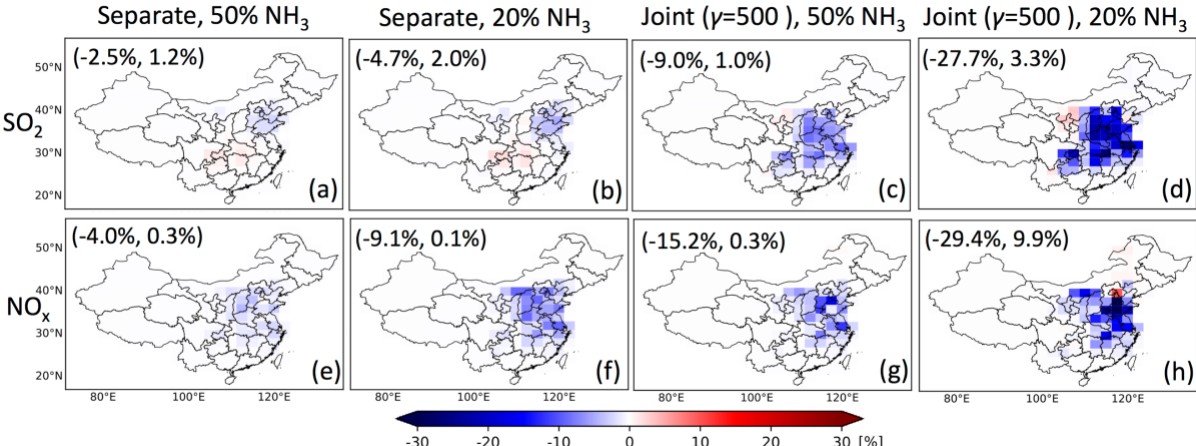


**Figure 9. Relative changes of posterior SO₂ (top row) and NOₓ (bottom row) emissions from the scenarios of perturbing NH₃ emissions with respect to that using original NH₃ emission inventory. (a) and (b) are relative changes of posterior SO₂ emissions from E-SO2-0.5NH₃ and E-SO2-0.2NH₃ with respect to that from E-SO2, respectively. (c) and (d) are relative changes of posterior SO₂ emissions from E-joint-0.5NH₃-γ500 and E-joint-0.2NH₃-γ500 with respect to that** 930 **from E-joint-dγ (γ=500), respectively. (e) and (f) are relative changes of posterior NOₓ emissions from E-NO2-0.5NH₃ and E-NO2-0.2NH₃ with respect to that from E-NO2, respectively. (g) and (h) are similar to (c) and (d), respectively, but for posterior NOₓ emissions. Minimum and maximum are shown in brackets.**

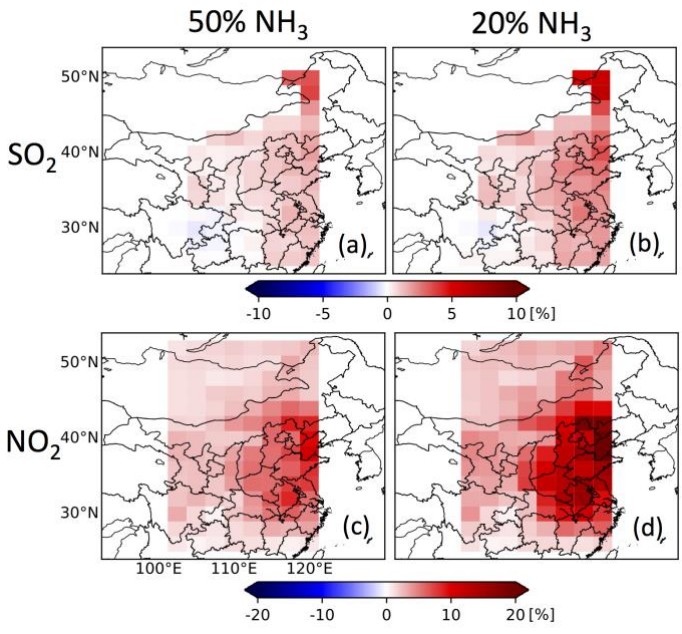


**Figure 10. Relative change of GEOS-Chem SO₂ VCDs when NH₃ emissions reduce to 50% (a) and 20% (b), respectively at OMPS overpassing time. (c) and (d) are similar to (a) and (b), respectively, but for NO₂.**

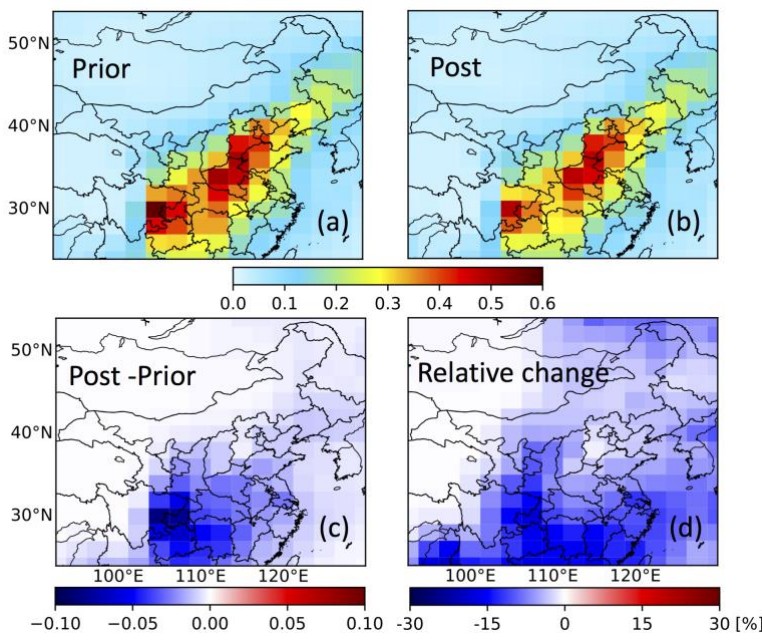


**Figure 11. Sulfate-nitrate-ammonium aerosol optical depth in prior (a) and posterior joint inversion (γ=500) (b). (c) is the difference between (b) and (a), and (d) is relative change in percentage.**

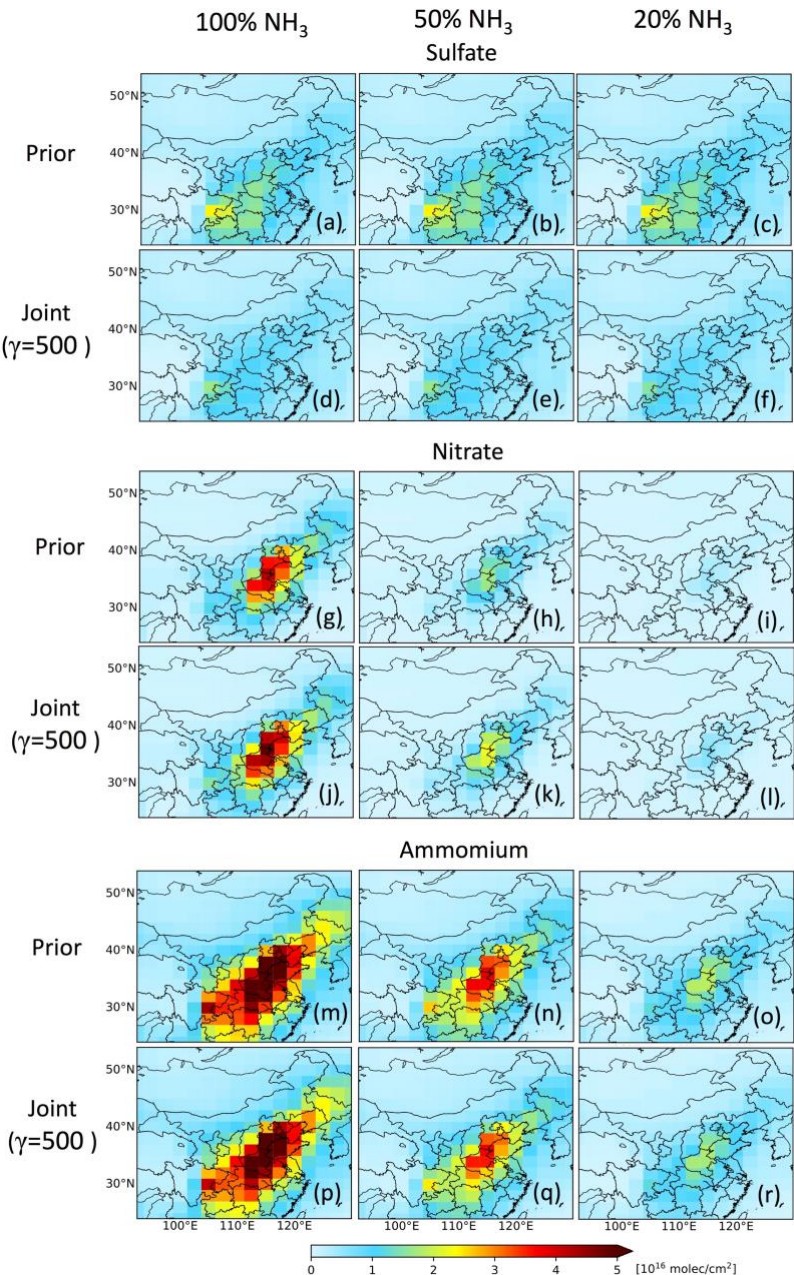

**Figure 12. Sulfate, nitrate, and ammonium column loadings in different scenarios. (a), (b), and (c) are prior sulfate at 100%, 50%, and 20% $NH_3$ emissions, respectively. (d), (e), and (f) are posterior sulfate from joint inversions (γ=500) at 100%, 50%, and 20% $NH_3$ emissions, respectively. (g)-(i) and (m)-(r) are similar to (a)-(f), but for nitrate and ammonium, respectively.**

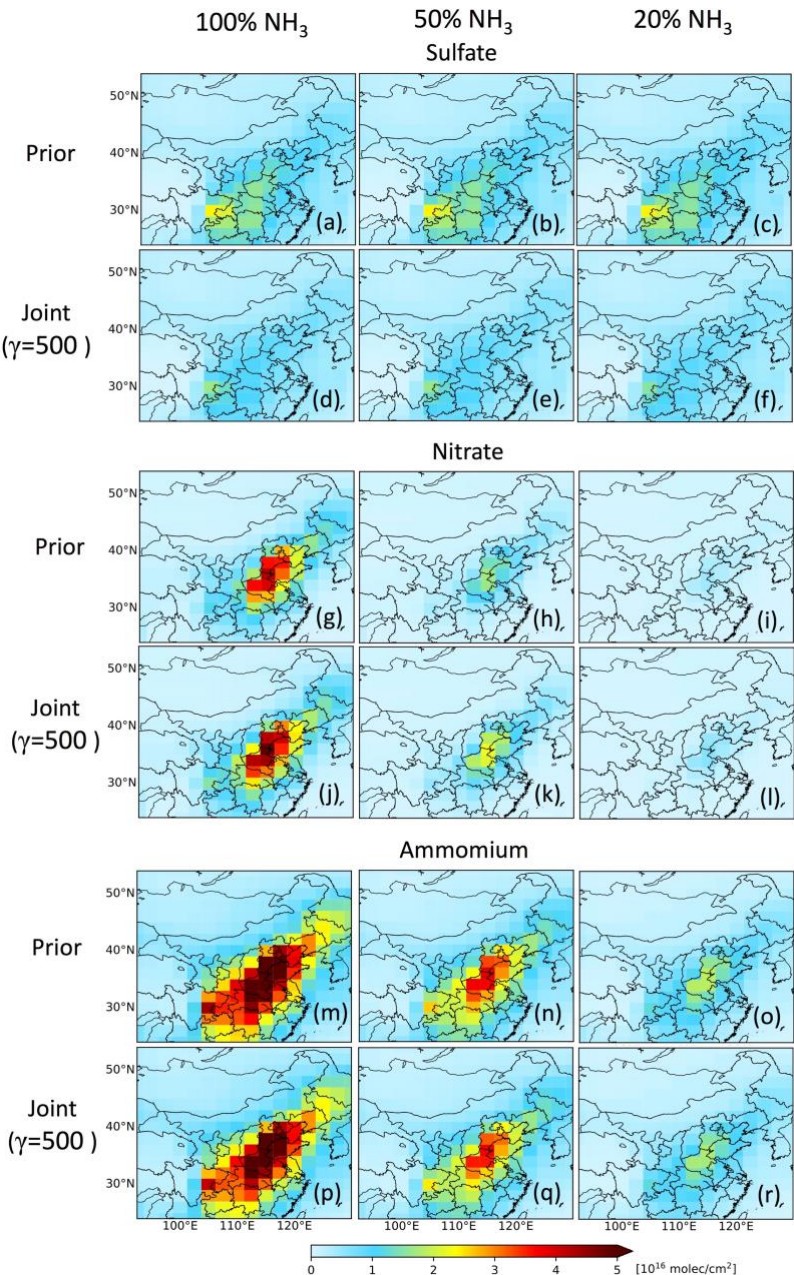