# Peer review of "Inverse modeling of SO2 and NOx emissions over China using multi-sensor satellite data: 1. formulation and sensitivity analysis"

_Atmospheric Chemistry and Physics, 2019_

## Referee Comment (RC1) · Anonymous Referee #1 · 11 Dec 2019

This paper presents the formulation and sensitivity analysis for the inverse modeling of SO2 and NOx emission over China using satellite data. While the authors seem to emphasize that the joint assimilation saves 50% of the computational time than assimilating SO2 and NO2 separately, the benefit of the joint assimilation should be more than that. This needs to be clarified.

In the paper, $\gamma$ is introduced to balance the SO2 and NO2 terms. In theory, it is not needed if the uncertainty terms can be well quantified. The optimal value of $\gamma$ is determined pretty arbitrarily. There are objective ways (such as Hollingworth-Lönnberg and NMC methods) to determine the observational errors and its covariance terms instead

of relying on arbitrary balancing.

Specific comments:

Line 215: It is surprising for the authors to choose less than 3% reduction in the cost function between two iterations as a criterion to halt the minimization. The L-BFGS-B can be very slow. Such a condition can often terminate the minimization prematurely. This needs to be changed if it is not a typo.

Lines 234-7: Is it really beneficial to balance the cost function this way? Can the SO2 observation errors be objectively determined?

Line 247: It is not accurate to say "emissions are adjusted mainly at locations where prior emissions are large". If there are non-zero emissions, the adjustments can be made. The limitation of using scaling factors is that zero-emission grid points cannot be modified.

Technical correction:

Line 161, "Terrain reflectivity less than $30^o$": Which angle does "Terrain reflectivity" refer to?

Line 171: GOES-FP -> GEOS-FP

Line 433: compared the latter -> compared with the latter
* * *

---

## Referee Comment (RC2) · Anonymous Referee #2 · 1 Jan 2020

This manuscript presents joint inversion results of SO2 and NOx emissions over China using the GEOS-Chem adjoint model and OMPS satellite observations for October 2013. The inversion results were compared against assimilated OMPS observations and independent OMI observations. Several sensitivity calculations were conducted to optimize the joint inversion framework. The joint inversion approach is unique, while the comparison against the OMI observations is interesting. I would, however, advise the authors to revise the manuscript. These revisions should be made before the manuscript can be considered for publication in ACP.

[ Major comments ]

[Figure]

The model horizontal resolution (2°x2.5° resolution) is clearly too coarse for current regional (not global) emission research, which could lead to serious problems for many applications (e.g., systematic biases in the downscaling analysis (Part 2)). In the previous study by the author's group (Qu et al, 2019), regional Chinese regional emissions were estimated at 0.5°x0.667° resolution using a hybrid 4D-Var/Mass balance approach to save computational resources for the multiple-year calculations, while conducting a one-month adjoint calculation at 0.5°x0.667° resolution using the same adjoint model with a nested domain for East Asia. In the same way, one-month inversion calculation at 0.5°x0.667° resolution using OMPS observations must be doable and should be tested in the present study. This is essential for evaluating the joint inversion performance using in-situ observations (please see my comment below), as already performed by Qu et al. (2019) for OMI assimilation results. It could also provide improved information (e.g., reduced systematic errors for each grid point, considering the non-linear chemistry) for down-scaling analysis (Part 2). For long-term emission estimations, the authors could still use the hybrid inversion framework at 0.5°x0.667° resolution (together with the downscaling approaches, if resolutions higher than at 0.5°x0.667° resolution are needed). Thus, I don't think the coarse resolution regional joint inversion will be needed for any applications. At the very least, 0.5°x0.667° resolution joint inversion calculations should be performed for key experiments.

The joint inversion results, including those from the sensitivity calculations, need to be evaluated against independent in-situ measurements, in order to obtain the optimized system. For this, the authors need to use their 0.5°x0.667° resolution joint inversion system. Resolutions higher than 0.5°x0.667° would be required for reducing representation gaps, as discussed in Part 2. Nevertheless, Qu et al (2019) already demonstrated that joint inversions at 0.5°x0.667° resolution can be evaluated using in-situ surface observations. This is also essential for evaluating possible biases in both OMPS and OMI satellite observations, which can be one of the most important results from the present study.

Although the joint inversion reduced the total computational cost, its scientific benefits (required for ACP, not for GMD) are not very clear. The discussions in Sections 4.4 and 4.5 are interesting. Adding evaluations using any AOD, NH3, and relevant observations would be helpful to demonstrate the scientific value of the joint inversion.

[ A few more specific comments ]

L203 "In this study, OMPS SO2 and NO2 tropospheric VCDs are retrieved using the shape of NO2 vertical profiles from GEOS-Chem simulations (Yang et al., 2013; Yang et al., 2014), although differences of model version, simulation year, and emission inventory still exists" These profiles can be largely different. The lack of averaging kernel in the observation operator can lead to serious problems. Please justify and demonstrate its impacts. Otherwise, data assimilation adjustments can be meaningless.

L204 "Hence, the difference between the GEOS-Chem simulations and the OMPS retrievals is mostly ascribed to the uncertainty of the emissions." This may not be true and requires further investigation.

OMI L3 data is used for validation. Without applying the averaging kernels, comparisons may not provide meaningful information. This needs to be investigated.

L325 and some other paces, "Our finding of a large reduction..." The discussion about trends between the 2013 October inversion and the 2010 inventories does not make any sense.

L330 "in some model grid cells": Please discuss the spatial pattern.

Section 4.2 does not provide very useful information and can be removed or shortened.
* * *

---

## Author Comment (AC1) · 14 Mar 2020

**Reply to reviewers and editors:**

We thank all reviewers for their careful reading of the manuscript, and for their many constructive feedbacks. The original comments by reviewers are in black font, our replies are in blue.

**Reviewer #1**

This paper presents the formulation and sensitivity analysis for the inverse modeling of SO2 and NOx emission over China using satellite data. While the authors seem to emphasize that the joint assimilation saves 50% of the computational time than assimilating SO2 and NO2 separately, the benefit of the joint assimilation should be more than that. This needs to be clarified.

Thanks for your comment. Yes, joint inversion should have more benefits in addition to save computational time. We have added the discussion below to Sect. 4.3.

When evaluating with OMI retrievals, joint inversion shows better results than separate inversion for $SO_2$ or $NO_2$, but not both, depending on the value of $\gamma$. When $\gamma$ is 20, 50, or 100, $NO_2$ NCRSME for E-joint-d$\gamma$ improvement appears to be smaller than that for E-$NO_2$, but $SO_2$ NCRSME for E-joint-d$\gamma$ is larger than that for E-$SO_2$. Conversely, when $\gamma$ is 1000, 1500, or 2000, $SO_2$ NCRSME for E-joint-d$\gamma$ is smaller than that for E-$SO_2$, but $NO_2$ NCRSME for E-joint-d$\gamma$ is larger than that for E-$NO_2$. This is similar to the findings by Qu et al., (2019b) in which the months when joint inversion show better result than separate inversion for $SO_2$ ($NO_2$) have worse result for $NO_2$ ($SO_2$). The benefit of joint inversion for improving only one species is similar to Qu et al. (2019b) and is likely to due to the complicated relationship between these two species through different chemical pathways. For example, $O_3$ and OH are key species that connect the chemistry of $SO_2$ and $NO_2$ and aerosols can affect the photolysis and heterogenous chemistry. Hence, while joint inversion to improve both species can not be demonstrated here, it should be reviewed as the first step of simultaneously assimilating multiple species (including AOD, $NH_3$, and other trace gases) to optimize emissions. Until then, the system is not ready to holistically evaluate the benefits of joint assimilation to improve the model in a systematic manner. It is worthy noting that Xu et al. (2013) showed the feasibility of using MODIS cloudfree radiance to optimize emissions of $SO_2$ and $NO_2$ at the same time. Future research should add the aerosol optical depth or visible reflectance (as well as tropospheric $O_3$ if reliable) as constraints to further evaluate the benefits of joint assimilation for improving model overall performance in a systematic matter.

In the paper, $\gamma$ is introduced to balance the SO2 and NO2 terms. In theory, it is not needed if the uncertainty terms can be well quantified. The optimal value of $\gamma$ is determined pretty arbitrarily. There are objective ways (such as Hollingworth-Lönnberg and NMC methods) to determine the observational errors and its covariance terms instead of relying on arbitrary balancing.

Thank you for the comments. In a practical way, we still need use $\gamma$ to balance the $SO_2$ and $NO_2$ terms no matter how observational errors are quantified, as "When it is not balanced, $SO_2$ observations have very little impact on the inversion results as the optimization algorithm will firstly minimize the observational term for $NO_2$ unless many more iterations than is computationally feasible are performed, which is caused by the fact that observational error and valid number of $NO_2$ observation are respectively smaller and larger than the counterparts of $SO_2$. We thus **subjectively** derive $\gamma$ in a **non-arbitrary** way in order to focus equally on both species, thereby tackling the imbalance in their observational constraints. In this manner, the cost function is defined to server the purpose of joint inversion of $SO_2$ and $NO_2$ emissions" "Similar balance approach that adjusts contribution of observation terms in the cost function is used in the past work that assimilates satellite trace gas retrievals to invert emissions (Qu et al., 2019b) or invert the aerosol optical properties from skylight polarization measurements of AERONET (Xu et al., 2015).." We have added the quoted text to Sect. 3.2.1.

Actually, OMPS $SO_2$ and $NO_2$ observational errors are quantified in an **objective** way. In a clean region (such as equatorial Pacific ocean (10°S–10°N,120°W–150°W ) that is far from emission sources), true $SO_2$ ($NO_2$) concentrations should be zero or negligible, while both negative and positive retrieval values exist. Thus, it is reasonable to use the variance of $SO_2$ ($NO_2$) retrievals over the clean region to represents $SO_2$ ($NO_2$) observation error variance. This error estimation approach is widely used in trace gas retrieval research community (Li et al.,

2013;Yang et al., 2013). These precision values can be used as the observation error in the cost function of data assimilation. However, we should notice that the estimated observation (retrieval) errors only represent the observation error distribution of the products a whole; it cannot represent the observation error distribution for every pixel, because the pixel-level error is amenable to spatiotemporal change of cloud fraction, satellite observation geometry, aerosol impacts, etc. In theory, if the uncertainties can be analytically described at the pixel level, they would be directly applied to improve the satellite product in the first place.

We have add the discussion above in Sect. 2.1.

We also have added the text below in Sect. 3.1

In the optimization formulation, the forward model errors are also considered as part of the observation error term.  However, while several ways to construct model error covariance matrix exist, including the Hollingworth-Lönnberg (Hollingsworth & Lönnberg, 1986) and NMC (Bannister, 2008) methods, their application for off-line CTM model error characterization deserves a separate a study. The Hollingsworth method extracts observation error variance (including forward model error) from (observation – background) covariance statistics with the assumptions that observation error is spatially uncorrelated, background error is spatially correlated as a function of distance, and observation error and background error are uncorrelated. The assumption that background error is spatially correlated as a function of distance only is suitable for the meteorological fields that vary smoothly, but for chemical species, emissions also contribute significantly to model errors and emissions are spatially correlated. The NMC method is normally applied to weather forecast models or on-line-coupled weather-chemistry models (Benedetti and Fisher, 2007). Off-line CTMs such as GEOS-Chem use the meteorological reanalysis and so, NMC is not applicable here to quantify the CTM's transport error. Consequently, CTM's transport errors are neglected in the past emission optimization work (Wang et al., 2016) and are adopted in this study. Admittedly, this simplification should be studied in future together with the evaluation and developments of methods to characterize off-line CTM errors.

**Specific comments:**

Line 215: It is surprising for the authors to choose less than 3% reduction in the cost function between two iterations as a criterion to halt the minimization. The L-BFGS-B can be very slow. Such a condition can often terminate the minimization prematurely. This needs to be changed if it is not a typo.

This is a good point. L-BFGS-B can be very slow to converge. But at some point we run into practical limitations on the amount of time we can spend iterating, thus we choose less than 3% reduction in the cost function between two successful iterations as a criterion to halt the minimization. Based on the criterion, for E-SO$_2$ and E-NO$_2$, we picked 5$^{th}$ and 6$^{th}$ iteration result, respectively. Further tests show that the more iterations (after <3% reduction of cost function) doesn't yield discernable difference in the cost function values (Fig. S3) and optimization results (Table S1 and S2). We have added the figure and table below to the supplement and added corresponding description at last paragraph of Sect. 3.1.

[Figure]

**Figure S3. Normalized cost function (the ratio of the cost function at a iteration to that at the 1$^{st}$ iteration) for E-SO$_2$ (a) and E-NO$_2$ (b). A iteration is accepted (solid circle) if its cost function value is smaller than that of any previous iterations, otherwise not accepted (empty circle). The 1$^{st}$ iteration (prior) is defined as not accepted. The iterations that are selected based on the halt criterion are marked with red cross.**

Table S1. Anthropogenic $SO_2$ emissions for October 2013 from E-$SO_2$ at each iteration.

| Iteration | 1 (prior) | 5 (selected) | 6 | 7 | 8 | 9 | 10 | 11 |
|---|---|---|---|---|---|---|---|---|
| $SO_2$ [Gg S] | 1166 | 748 | 744 | Not accepted[a] | 746 | Not accepted | 749 | Not accepted |

[a]Posterior emission total amount at the iteration that is not accepted (cost function value is not smaller than that of any previous iterations) is not shown.

Table S2. Anthropogenic $NO_x$ emissions for October 2013 from E-$NO_2$ at each iteration.

| Iteration | 1 (prior) | 6 (selected) | 7 | 8 | 9 | 10 | 11 | 12 |
|---|---|---|---|---|---|---|---|---|
| $NO_x$ [Gg N] | 714 | 672 | 667 | Not accepted[a] | 666 | 666 | 666 | Not accepted |

[a]Posterior emission total amount at the iteration that is not accepted (cost function value is not smaller than that of any previous iterations) is not shown.

Lines 234-7: Is it really beneficial to balance the cost function this way? Can the SO2 observation errors be objectively determined?

As we address that comment about using $\gamma$ to balance observation term, $SO_2$ observation error is **objectively** determined, but the spatial balance problem still exists. Similar balance approach that adjusts contribution of observation terms in the cost function is used in the past work that assimilates satellite trace gas retrievals to invert emissions (Qu et al., 2019b) or invert the aerosol optical properties from skylight polarization measurements of AErosol RObotic NETwork (AERONET) (Xu et al., 2015). Thus we think the subjective but non-arbitrary balance approach should be acceptable, although it is a break from strict Bayesian derivation of the cost function. We have added how the $SO_2$ observation error (0.2 DU) are determined in an objective way in Sect. 2.1 and the justification of the balance approach in Sect. 3.2.1.

Line 247: It is not accurate to say "emissions are adjusted mainly at locations where prior emissions are large". If there are non-zero emissions, the adjustments can be made. The limitation of using scaling factors is that zero-emission grid points cannot be modified.

Yes. We agree that "The limitation of using scaling factors is that zero-emission grid points cannot be modified." We have added statement of the limitation in the manuscript.

***Technical correction:***

Line 161, "Terrain reflectivity less than 30o": Which angle does "Terrain reflectivity" refer to?

This is a typo. We have change it to "terrain reflectivity less than 0.3".

Line 171: GOES-FP -> GEOS-FP

Corrected.

Line 433: compared the latter -> compared with the latter

Corrected.

---

## Author Comment (AC2) · 14 Mar 2020

**Reply to reviewers and editors:**

We thank all reviewers for their careful reading of the manuscript, and for their many constructive feedbacks. The original comments by reviewers are in black font, our replies are in blue.

**Reviewer #2**

This manuscript presents joint inversion results of SO2 and NOx emissions over China using the GEOS-Chem adjoint model and OMPS satellite observations for October 2013. The inversion results were compared against assimilated OMPS observations and independent OMI observations. Several sensitivity calculations were conducted to optimize the joint inversion framework. The joint inversion approach is unique, while the comparison against the OMI observations is interesting. I would, however, advise the authors to revise the manuscript. These revisions should be made before the manuscript can be considered for publication in ACP. Thanks for the positive comments. We did our best to address them in the revision.

*[ Major comments ]*

The model horizontal resolution (2°x2.5° resolution) is clearly too coarse for current regional (not global) emission research, which could lead to serious problems for many applications (e.g., systematic biases in the downscaling analysis (Part 2)). In the previous study by the author's group (Qu et al, 2019), regional Chinese regional emissions were estimated at 0.5°x0.667° resolution using a hybrid 4D-Var/Mass balance approach to save computational resources for the multiple-year calculations, while conducting a one-month adjoint calculation at 0.5°x0.667° resolution using the same adjoint model with a nested domain for East Asia. In the same way, one-month inversion calculation at 0.5°x0.667° resolution using OMPS observations must be doable and should be tested in the present study. This is essential for evaluating the joint inversion performance using in-situ observations (please see my comment below), as already performed by Qu et al. (2019) for OMI assimilation results. It could also provide improved information (e.g., reduced systematic errors for each grid point, considering the non-linear chemistry) for down-scaling analysis (Part 2). For long-term emission estimations, the authors could still use the hybrid inversion framework at 0.5°x0.667° resolution (together with the

downscaling approaches, if resolutions higher than at 0.5◦x0.667◦ resolution are needed). Thus, I don't think the coarse resolution regional joint inversion will be needed for any applications. At the very least, 0.5◦x0.667◦ resolution joint inversion calculations should be performed for key experiments.

Thanks for the good suggestions. We acknowledge that it is better to optimize emissions directly at fine resolution (such as 0.5°x0.667°, and 0.25°x0.3125°) rather than coarse resolution (such as 2°x2.5°), and that it is doable for one-month inversion at fine resolution. After careful consideration, we think it is both practical and reasonable to assimilate OMPS retrievals at the resolution of 2°x2.5°, as OMPS pixel size could be much larger than fine-resolution grid boxes (and OMI whose pixel size is 13 km x 24 km at nadir and 26 km x 128km at edge). OMPS pixel size is 50 km x 50 km at nadir, and becomes 190 km x 50 km at edges. Thus, OMPS pixel size is comparable to (at nadir) or much larger than (at edges) 0.5°x0.667° grid box. An OMPS pixel may cover several 0.5°x0.667° grid boxes and cannot resolve variations of concentrations owing to variations emissions at that fine resolution. Currently, the nested GOES-Chem adjoint model only supports the 0.5°x0.667° GEOS-5 meteorological field and the 0.25°x0.3125° GEOS-FP meteorological field. And the GEOS-5 meteorological filed has a temporal coverage from 2004 to mid-2013; data after mid-2013, e.g., our study time period, is unavailable. Although 0.25°x0.3125° GEOS-FP meteorological field is up to date, apparently its resolution is too fine to compare with OMPS. We have added this explanation in Sect. 3.2.

The joint inversion results, including those from the sensitivity calculations, need to be evaluated against independent in-situ measurements, in order to obtain the optimized system. For this, the authors need to use their 0.5◦x0.667◦ resolution joint inversion system. Resolutions higher than 0.5◦x0.667◦ would be required for reducing representation gaps, as discussed in Part 2. Nevertheless, Qu et al (2019) already demonstrated that joint inversions at 0.5◦x0.667◦ resolution can be evaluated using in- situ surface observations. This is also essential for evaluating possible biases in both OMPS and OMI satellite observations, which can be one of the most important results from the present study.

As answered the previous question, due to large pixel size of OMPS, we didn't attempt to constrain the emissions at the resolution finer than the satellite instrument on the monthly basis. In addition, for those retrievals at high resolution (such as TROPOMI), the retrieval uncertainty is expected to

be alleviated after aggregating pixel-level retrieval into the coarser resolution. Furthermore, even if the inversion were conducted at the resolution of 0.5°x0.667° or 0.25°x0.3125°, it is still very challenging to evaluating possible bias in both OMPS and OMI satellite observations, as representation gaps still exists in the two resolutions. Zheng et al. (2017) showed that surface $SO_2$ ($NO_2$) concentration simulations from WRF-CMAQ, when evaluating with in situ observations, have a NMB of -23% (%0), 7% (32%), and 41% (45%) at the resolutions of 36 km (~0.36°), 12 km (~0.12°), and 4 km (~0.04°), respectively; this shows that representation gaps still exist at 12 km (~0.12°), which is already finer than 0.5° and 0.25°. Thus, perhaps it is not surprising that large negative bias exists when evaluating posterior GEOS-Chem 0.5°x0.667° simulations with in situ $SO_2$ (Fig. 10 in Qu et al. (2019)) and $NO_2$ (Fig. 11 in Qu et al. (2019)) in Qu et al. (2019), and it is somewhat assertive to conclude that the negative bias imply the negative bias of OMI $SO_2$ and $NO_2$ retrievals. In Qu et al. (2019), the improvements of posterior simulations when evaluating with in situ $SO_2$ and $NO_2$ surface concentrations are mainly represented by Normalized Mean Square Error (NMSE) rather than bias.

The goal of this paper is NOT to replicate the method by Qu et al. (2019). Rather, the goal of this paper is to illustrate how OMPS data could be used to improve an air quality *forecast* through monhtly update of emissions (possibly in near real time manner) at a resolution much finer than OMPS. Hence, if implemented, our method of using 2°x2.5° resolution for performing the optimization can save considerable computational time (and is much more feasible for a research group such us in the university), and then using the downscaling method (part II developed by this study), the finer resolution forecast can be made in a practical manner (suitable for a regional modeling group for air quality forecast). In contrast, the focus of Qu et al (2019) is the re-analysis of emissions, as opposed to forecasting of air quality at the finer scales. In contrast, optimization at 0.5° x 0.67° will still require downscaling method for air quality forecast (normally at ~10 km resolution). We have added the elaborations above in section 3.2.

Although the joint inversion reduced the total computational cost, its scientific benefits (required for ACP, not for GMD) are not very clear. The discussions in Sections 4.4 and 4.5 are

interesting. Adding evaluations using any AOD, NH3, and relevant observations would be helpful to demonstrate the scientific value of the joint inversion.

We have expanded discussion of the scientific benefits of the approach in response to the first comment from the reviewer 1, see above. Further, ACP's scope is very broad and developing data assimilation techniques for using new satellite data has been published in ACP, such as Chen et. al (2018). Please look at this part I and part II paper as a whole – they effectively showcase an approach that is economic in computation to use OMPS data to improve air quality forecast at fine scale. In part II, we did many independent evaluations. Furthermore, we also studied the results of the sensitivity to $NH_3$ – a topic that has not been studied before in data assimilation. We consider they have good scientific merits.

*[ A few more specific comments ]*

L203 "In this study, OMPS SO2 and NO2 tropospheric VCDs are retrieved using the shape of NO2 vertical profiles from GEOS-Chem simulations (Yang et al., 2013; Yang et al., 2014), although differences of model version, simulation year, and emission inventory still exists" These profiles can be largely different. The lack of averaging kernel in the observation operator can lead to serious problems. Please justify and demonstrate its impacts. Otherwise, data assimilation adjustments can be meaningless.

Thanks for pointing out this. We totally agree that differences of model version, simulation year, and emission inventory could lead to profile differences. Thus, we compare operational OMPS retrievals with VCDs modified through averaging kernel to investigate how much VCD differences are caused by profile differences. We have added the discussion below to Sect. 4.1.1 and figures below to supplement.

For $SO_2$

The $SO_2$ NMB (106.5%) between GOES-Chem prior simulation and OMPS is much larger than the NMB (-6.8%, Fig S1) caused by the difference of $SO_2$ vertical profiles between OMPS $SO_2$ retrieval algorithm and current prior simulation; thus averaging kernel is not considered in the OMPS $SO_2$ observation operator.

[Figure]

**Figure S1. OMPS SO₂ Vertical Column Density (VCD) retrievals in Arpil 2018.** (a) and (b) are operational VCDs and the VCDs that are modified through averaging kernel according to formula S1, respectively. (c) is the differences between the modified and operational VCDs. (d) is scatter plot of modified VCDs versus operational VCDs. Linear correlation coefficient (R), linear regression equation, root mean squared error (RMSE), normalized mean bias (NMB), mean bias (MB), and number of observations (N) are shown over the scatter plot.

For NO₂

Similarly, the averaging kernel is not considered in the OMPS NO₂ observation operator for optimization for the following reasons. First, the OMPS NO₂ retrieval differences due to the profile differences can lead to a NMB of -7.5% (Fig S2), which is still smaller than the prior GEOS-Chem simulation NMB (10.9%, Fig. 3l). Second, a NMB of 10.9% for model NO₂ VCD simulation is not a very large value, as the difference between satellite NO₂ VCD retrievals and ground-based measurements could be comparable to this value. For example, Krotkov et al. (2017) shows that OMI NO₂ VCD retrievals, on average, are ~10% larger than ground-based FTIR spectrometer. Thus, current research should mainly focus on the change of the spatial distribution (such as linear correlation coefficient) rather than bias of prior and posterior GEOS-Chem NO₂ VCD simulation. Finally, given that linear correlation coefficient between OMPS retrievals and that are modified through integration of averaging kernel and NO₂ vertical profile from this study is as large as 0.99, averaging kernel is not treated in the OMPS NO₂ observation operator.

[Figure]

**Figure S2. OMPS NO₂ Vertical Column Density (VCD) retrievals in October 2013. (a) and (b) are operational VCDs and the VCDs that are modified through averaging kernel according to formula S1, respectively. (c) is the differences between the modified and operational VCDs. (d) is scatter plot of modified VCDs versus operational VCDs. Linear correlation coefficient (R), linear regression equation, root mean squared error (RMSE), normalized mean bias (NMB), mean bias (MB), and number of observations (N) are shown over the scatter plot.**

L204 "Hence, the difference between the GEOS-Chem simulations and the OMPS retrievals is mostly ascribed to the uncertainty of the emissions." This may not be true and requires further investigation.

We acknowledge that when averaging kernel is not considered in the observation operator, the profile differences can contribute to the difference between the GEOS-Chem simulations and the OMPS retrievals. Additionally, GEOS-Chem model uncertainty can also contribute to the difference between the GEOS-Chem simulations and the OMPS retrievals, and it is difficult to estimate model uncertainty. Thus, we delete the sentence in the revised manuscript; we added the discussion in Sect. 4.1.1 to show that difference between the GEOS-Chem simulations and the operational OMPS retrievals is larger than difference between operational OMPS retrievals and retrievals modified through consideration of averaging kernel. Therefore, the inverse modeling results are statistically significant. We also acknowledge that GEOS-Chem model uncertainty affects inverse modeling results, thus we apply optimized emission inventory to another model (different version of the GEOS-Chem model with much finer resolution) to show that improvement of air quality simulation and forecasts is obtained though the uncertainty of

models. In addition, we also follow the suggestion of evaluating the optimized emission inventory in a consistent framework for part II manuscript; please see details in our reply for part II manuscript.

OMI L3 data is used for validation. Without applying the averaging kernels, comparisons may not provide meaningful information. This needs to be investigated.

Thanks for the suggestion. We acknowledge there should be differences between apply and do not apply averaging kernels (or scattering weights) in the comparisons. Following the suggestion, we investigate how this affect evaluation. In the manuscript, OMI L3 $SO_2$ and OMI L3 $NO_2$ are used for evaluation. In OMI L3 $SO_2$ dataset, only the best pixel in a 0.25°x0.25° grid cell is retained, and the observational geometry information for the pixel is also available. Thus, we can still apply scattering weights to OMI L3 $SO_2$. In OMI L3 $NO_2$ dataset, observational geometry information it not available, thus we can only apply scattering weights to OMI L2 $NO_2$. The text and figures below are added to supplement. We have emphasized in the main text that these conclusions do not change in Sect. 4.1.3.

[Figure]

Figure S4. Taylor diagrams for comparing of VCDs of SO₂ (a) and NO₂ (b) from the GEOS-Chem simulations (squares for prior, triangles for posterior E-SO₂ (a) or E-NO₂ (b), and diamonds for E-joint) with that from the OMI (label 1 for operational level 3 SO₂ (a) or level 3 NO₂ (b) and label 2 for the level 3 SO₂ (a) that are modified by considering the vertical profiles from the GEOS-Chem simulation with which is to be compared or the level 2 NO₂ (b) that are modified by considering the vertical profiles from the GEOS-Chem simulation with which is to be compared) in October 2013 over China.

$$\Omega_s^{opl} = \Omega_v^{opl} \cdot M^{opl} \quad \text{(S2)}$$

$$M^{new} = W \cdot S^{new} \quad \text{(S3)}$$

$$\Omega_v^{opl} = \frac{\Omega_s^{opl}}{M^{new}} \quad \text{(S4)}$$

Equation S2 is used to convert OMI SO₂ (NO₂) vertical column density $\Omega_v^{opl}$ to SO₂ (NO₂) slant column density $\Omega_s^{opl}$ by multiplying SO₂ (NO₂) air mass factor $M^{opl}$ from OMI product. Equation S3 is used to calculate new SO₂ (NO₂) air mass factor $M^{new}$, where $W$ is SO₂ (NO₂) scattering weight, and $S^{new}$ is SO₂ (NO₂) shape factor that from the GEOS-Chem simulation with which is to be compared. Equation S4 is used to calculate new OMI SO₂ (NO₂) vertical column density $\Omega_v^{opl}$.

Whether use OMI data without applying scattering weight (Label 1 in Fig. S4) or OMI data with applying scattering weight (Label 2 in Fig. S4), the main conclusions in Sect. 4.1.3 does not change. These conclusions are:

(1) For SO$_2$, posterior GEOS-Chem simulations (E-SO$_2$ and E-joint) show smaller NMB and better spatial distribution (in terms of NCRMSE) than prior GEOS-Chem simulations when evaluating with OMI SO$_2$ (apply or not apply scattering weight).

(2) For NO$_2$, the improvements for E-NO$_2$ and E-joint are reflected in terms of R when evaluating with OMI tropospheric VCDs (apply or not apply scattering weight), although the two experiments show larger negative NMB than the prior simulation.

L325 and some other paces, "Our finding of a large reduction. . ." The discussion about trends between the 2013 October inversion and the 2010 inventories does not make any sense.

Thanks for pointing out this. The MIX 2010 inventory was derived through bottom-up approach, while the 2013 October inversion inventory is derived through integration of GEOS-Chem adjoint model and OMPS SO$_2$ and NO$_2$ vertical column density retrievals. We have to acknowledge that systematic bias exists in both of the inventories, and so, the difference between the two emission inventories should not be considered as trend. To investigate trends, emission inventories should be derived from the same approach. In the revision, we therefore have removed the discussions and emphasized that the differences should not be considered as trends.

L330 "in some model grid cells": Please discuss the spatial pattern.

Sure. We have added "Although the relative difference between E-joint and E-NO$_2$ proved to be less than 2% in terms of total anthropogenic NO$_x$ emissions over China (Fig. 4k), it is up to 40% over Shanxi province, and both grids with large positive differences and grids with large negative differences exist over North China Plain (Fig. 4 l)." in section 4.1.2.

Section 4.2 does not provide very useful information and can be removed or shortened.

Thank you for the comment. This section shows the impacts of data quality control and spatial balance. Although it does not include much scientific information, it helps to understand how observational errors are assumed. Thus, we hope to keep it.

**Reference**

Bannister, R. N.: A review of forecast error covariance statistics in atmospheric variational data assimilation. I: Characteristics and measurements of forecast error covariances, Quarterly Journal of the Royal Meteorological Society, 134, 1951-1970, 10.1002/qj.339, 2008.

Chen, C., Dubovik, O., Henze, D. K., Lapyonak, T., Chin, M., Ducos, F., Litvinov, P., Huang, X., and Li, L.: Retrieval of desert dust and carbonaceous aerosol emissions over Africa from POLDER/PARASOL products generated by the GRASP algorithm, Atmos. Chem. Phys., 18, 12551–12580, https://doi.org/10.5194/acp-18-12551-2018, 2018.

Hollingsworth, A., and Lönnberg, P.: The statistical structure of short-range forecast errors as determined from radiosonde data. Part I: The wind field, Tellus A, 38A, 111-136, 10.1111/j.1600-0870.1986.tb00460.x, 1986.

Krotkov, N. A., Lamsal, L. N., Celarier, E. A., Swartz, W. H., Marchenko, S. V., Bucsela, E. J., Chan, K. L., Wenig, M., and Zara, M.: The version 3 OMI NO2 standard product, Atmos. Meas. Tech., 10, 3133-3149, 10.5194/amt-10-3133-2017, 2017.

Li, C., Joiner, J., Krotkov, N. A., and Bhartia, P. K.: A fast and sensitive new satellite $SO_2$ retrieval algorithm based on principal component analysis: Application to the ozone monitoring instrument, Geophys. Res. Lett., 40, 6314-6318, 10.1002/2013GL058134, 2013.

Qu, Z., Henze, D. K., Theys, N., Wang, J., and Wang, W.: Hybrid Mass Balance/4D-Var Joint Inversion of NOx and SO2 Emissions in East Asia, J. Geophys. Res., 124, 8203-8224, 10.1029/2018JD030240, 2019.

Xu, X., Wang, J., Henze, D. K., Qu, W., and Kopacz, M.: Constraints on aerosol sources using GEOS-Chem adjoint and MODIS radiances, and evaluation with multisensor (OMI, MISR) data, J. Geophys. Res., 118, 6396-6413, 10.1002/jgrd.50515, 2013.

Xu, X., Wang, J., Zeng, J., Spurr, R., Liu, X., Dubovik, O., Li, L., Li, Z., Mishchenko, M. I., Siniuk, A., and Holben, B. N.: Retrieval of aerosol microphysical properties from AERONET photopolarimetric measurements: 2. A new research algorithm and case demonstration, J. Geophys. Res., 120, 2015JD023113, 10.1002/2015JD023113, 2015.

Yang, K., Dickerson, R. R., Carn, S. A., Ge, C., and Wang, J.: First observations of SO2 from the satellite Suomi NPP OMPS: Widespread air pollution events over China, Geophys. Res. Lett., 40, 4957-4962, 10.1002/grl.50952, 2013.

Zheng, B., Zhang, Q., Tong, D., Chen, C., Hong, C., Li, M., Geng, G., Lei, Y., Huo, H., and He, K.: Resolution dependence of uncertainties in gridded emission inventories: a case study in Hebei, China, Atmos. Chem. Phys., 17, 921-933, 10.5194/acp-17-921-2017, 2017.